# Learning-to-learn non-convex piecewise-Lipschitz functions

**Maria-Florina Balcan, Mikhail Khodak, Dravyansh Sharma**
Carnegie Mellon University
{ninamf,mkhodak,dravyans}@cs.cmu.edu

**Ameet Talwalkar**
Carnegie Mellon University & Hewlett Packard Enterprise
talwalkar@cmu.edu

## Abstract

We analyze the meta-learning of the initialization and step-size of learning algorithms for piecewise-Lipschitz functions, a non-convex setting with applications to both machine learning and algorithms. Starting from recent regret bounds for the exponential forecaster on losses with dispersed discontinuities, we generalize them to be initialization-dependent and then use this result to propose a practical meta-learning procedure that learns both the initialization and the step-size of the algorithm from multiple online learning tasks. Asymptotically, we guarantee that the average regret across tasks scales with a natural notion of task-similarity that measures the amount of overlap between near-optimal regions of different tasks. Finally, we instantiate the method and its guarantee in two important settings: robust meta-learning and multi-task data-driven algorithm design.

## 1 Introduction

While learning-to-learn or *meta-learning* has long been an object of study [45], in recent years it has gained significant attention as a multi-task paradigm for developing algorithms for learning in dynamic environments, from multiple data sources, and in federated settings. Such methods focus on using data from multiple tasks to improve performance when facing a new, potentially related task. A popular approach is *initialization-based* meta-learning, in which the meta-learner uses multi-task data to output an initialization for an iterative algorithm such as stochastic gradient descent (SGD) [25]. The flexibility of this approach has led to its widespread adoption, e.g. in robotics [23] and federated learning [18], and to a growing number of attempts to understand it, both empirically and theoretically [20, 29, 24, 40, 42]. However, outside some stylized setups our learning-theoretic understanding of how to meta-learn an initialization is largely restricted to the convex Lipschitz setting.

We relax both assumptions to study the meta-learning of online algorithms over piecewise-Lipschitz functions, which can be nonconvex and highly discontinuous. As no-regret online learning over such functions is impossible in-general, we study the case of piecewise-Lipschitz functions with *dispersed* discontinuities that do not concentrate in any small compact subset of the domain [11]. Such functions arise frequently in *data-driven algorithm design*, where the goal is to learn the optimal parameter settings of algorithms for difficult (often NP-Hard) problems over a distribution or sequence of instances [4]; for example, a small change to the metric in cluster linkage can lead to a discontinuous change in the classification error [7]. In this paper, we also demonstrate that such losses are relevant in the setting of adversarial robustness, where we introduce a novel online formulation. For both cases, the associated problems are often solved across many time periods or for many different problem domains, resulting in natural multi-task structure that might improve performance. To the best of our knowledge, ours is the first theoretical study of meta-learning in both of these application settings.

35th Conference on Neural Information Processing Systems (NeurIPS 2021).

In the single-task setting the problem of learning dispersed functions can be solved using simple methods such as the exponentially-weighted forecaster. To design an algorithm for learning to initialize online learners in this setting, we propose a method that optimizes a sequence of data-dependent upper-bounds on the within-task regret [29]. The result is an averaged bound that improves upon the regret of the single-task exponential forecaster so long as there exists an initial distribution that can compactly contain many of the within-task optima of the different tasks. Designing the meta-procedure is especially challenging in our setting because it involves online learning over a set of distributions on the domain. To handle this we study a "prescient" form of the classic follow-the-regularized leader (FTRL) scheme that is run over an unknown discretization; we then show the existence of another algorithm that plays the same actions but uses only known information, thus attaining the same regret while being practical to implement.

To demonstrate the usefulness of our method, we study this algorithm in two settings.

**Multi-task data-driven algorithm design.** We consider data-driven tuning of the parameters of combinatorial optimization algorithms for hard problems such as knapsack and clustering. The likely intractability of these problems on worst case instances have led to several approaches to study them in more realistic settings, such as smoothed analysis [44] and data-driven algorithm configuration [4]. Our setting is more realistic than those considered in prior work. It is more challenging than learning from iid instances [27, 12], but at the same time less pessimistic than online learning over adversarial problem instances [11] as it allows us to leverage similarity of problem instances coming from different but related distributions. We instantiate our bounds theoretically on several problems where the cost functions are piecewise-constant in the tuned parameters, allowing our meta-procedure to learn the right initial distribution for exponential forecasters. This includes well-known combinatorial optimization problems like finding the maximum weighted independent set (MWIS) of vertices on a graph, solving quadratic programs with integer constraints using algorithms based on the celebrated Goemans-Williamson algorithm, and mechanism design for combinatorial auctions. Then we consider experimentally the problem of tuning the right $\alpha$ for the $\alpha$-Lloyd's family of clustering algorithms [15]. In experimental evaluations on two datasets—a synthetic Gaussian mixture model and the well-known Omniglot dataset from meta-learning [33]—our meta-procedure leads to improved clustering accuracy compared to single-task learning to cluster. We also study our results for a family of greedy algorithms for the knapsack problem introduced by [27] and obtain similar results for a synthetic dataset. The results holds for both one-shot and five-shot tasks.

**Online robust meta-learning.** The second instantiation of our meta-learning procedure is to a new notion of adversarial robustness for the setting of online learning, where our results imply robust meta-learning in the presence of outliers. In this setting, the adversary can make (typically small) modifications to some example $x \in \mathcal{X}$, which can result in potentially large changes to the corresponding loss value $l_h(x)$, where $h \in \mathcal{H}$ is our hypothesis. For instance, consider the well-studied setting of adversarial examples for classification of images using deep neural networks [36, 17]. Given a neural network $f$, the adversary can perturb a datapoint $x$ to a point $x'$, say within a small $L_p$-ball around $x$, such that $f(x) = f(x')$ but the true label of $x'$ does not match $x$, and therefore $l_f(x) \neq l_f(x')$. In general, under the adversarial influence, we observe a *perturbed loss* function $\tilde{l}_h(x) = l_h(x) + a_h(x)$. Typically we are interested in optimizing both the perturbed loss $\tilde{l}_h(x)$, i.e. measuring performance relative to optimum for adversarially perturbed losses, and the *true loss* $l_h(x)$ (performance on the unobserved, unperturbed loss). For example, in the online learning setting, [1] consider perturbed loss minimization for linear dynamical systems, while [41] look at true $\{0, 1\}$ loss minimization in the presence of adversarial noise. Our approach ensures that regret for both the perturbed and true loss are small, for piecewise-Lipschitz but dispersed adversaries.

## 2 Preliminaries and initialization-dependent learning of dispersed functions

In this section we introduce our setup and notation for online learning of piecewise-Lipschitz functions in a multi-task environment. We then generalize existing results for the single-task setting in order to obtain within-task regret bounds that depend on both the initialization and the task data. This is critical for both defining a notion of task similarity and devising a meta-learning procedure.

**Meta-learning setup**  Following past setups [2, 21, 29], for some $T, m > 0$ and all $t \in [T]$ and $i \in [m]$ we consider a meta-learner faced with a sequence of $Tm$ loss functions $\ell_{t,i} : C \mapsto [0, 1]$ over a compact subset $C \subset \mathbb{R}^d$ that lies within a ball $\mathcal{B}(\rho, R)$ of radius $R$ around some point $\rho \in \mathbb{R}^d$.

---

**Algorithm 1** Exponential Forecaster

---

1: **Input:** step size parameter $\lambda \in (0, 1]$, initialization $w : C \to \mathbb{R}_{\geq 0}$.
2: Initialize $w_1 = w$
3: **for** $i = 1, 2, \ldots, m$ **do**
4: $\quad W_i := \int_C w_i(\rho) d\rho$
5: $\quad$ Sample $\rho_i$ with probability proportional to $w_i(\rho_i)$, i.e. with probability $p_i(\rho_i) = \frac{w_i(\rho_i)}{W_i}$
6: $\quad$ Suffer $\ell_i(\rho_i)$ and observe $\ell_i(\cdot)$
7: $\quad$ For each $\rho \in C$, set $w_{i+1}(\rho) = e^{-\lambda \ell_i(\rho)} w_i(\rho)$

---

Here we used the notation $[n] = \{1, \ldots, n\}$. Before each loss function $\ell_{t,i}$ the meta-learner must pick an element $\rho_{t,i} \in C$ before then suffering a loss or cost $\ell_{t,i}(\rho_{t,i})$. For a fixed $t$, the subsequence $\ell_{t,1}, \ldots, \ell_{t,m}$ defines a **task** for which we expect a single element $\rho_t^* \in C$ to do well, and thus we will use the **within-task regret** on task $t$ to describe the quantity

$$\mathbf{R}_{t,m} = \sum_{i=1}^{m} \ell_{t,i}(\rho_{t,i}) - \ell_{t,i}(\rho_t^*) \quad \text{where} \quad \rho_t^* \in \arg\min_{\rho \in C} \sum_{i=1}^{m} \ell_{t,i}(\rho) \tag{1}$$

In the single-task setting the goal is usually to show that $R_{t,m}$ is sublinear in $m$, i.e. that the average loss decreases with more rounds. A key point here is that the functions we consider can have numerous global optima. In this work we will assume, after going through the $m$ rounds of task $t$, that we have oracle access to a single fixed optimum for $t$, which we will refer to using $\rho_t^*$ and use in both our algorithm and to define the task-similarity. Note that in the types of applications we are interested in—piecewise-Lipschitz functions—the complexity of computing optima scales with the number of discontinuities. In the important special case of piecewise-constant functions, this dependency becomes logarithmic [19]. Thus this assumption does not affect the usefulness of the result.

Our goal will be to improve the guarantees for regret in the single-task case by using information obtained from solving multiple tasks. In particular, we expect average performance across tasks to improve as we see more tasks; to phrase this mathematically we define the **task-averaged regret**

$$\bar{\mathbf{R}}_{T,m} = \frac{1}{T} \sum_{t=1}^{T} \mathbf{R}_{t,m} = \frac{1}{T} \sum_{t=1}^{T} \sum_{i=1}^{m} \ell_{t,i}(\rho_{t,i}) - \ell_{t,i}(\rho_t^*) \tag{2}$$

and claim improvement over single-task learning if in the limit of $T \to \infty$ it is smaller than $\mathbf{R}_{t,m}$. Note that for simplicity in this work we assume all tasks have the same number of rounds within-task, but as with past work our results are straightforward to extend to the more general setting.

**Learning piecewise-Lipschitz functions** We now turn to our target functions and within-task algorithms for learning them: piecewise-Lipschitz losses, i.e. functions that are $L$-Lipschitz w.r.t. the Euclidean norm everywhere except on measure zero subsets of the space; here they may have arbitrary jump discontinuities so long they still bounded between $[0, 1]$. Apart from being a natural setting of interest due to its generality compared to past work on meta-learning, this class of functions has also been shown to have important applications in data-driven algorithm configuration [11]; there these functions represent the cost, e.g. an objective value or time-complexity, of algorithms for difficult problems such as integer programming, auction design, and clustering.

This literature has also shown lower bounds demonstrating that no-regret learning piecewise-Lipschitz function is impossible in general, necessitating assumptions about the sequence. One such condition is *dispersion*, which requires that the discontinuities are not too concentrated.

**Definition 2.1** ([11]). *The sequence of random loss functions $\ell_1, \ldots, \ell_m$ is said to be $\beta$-**dispersed** with Lipschitz constant $L$ if, for all $m$ and for all $\epsilon \geq m^{-\beta}$, we have that, in expectation over the randomness of the functions, at most $\tilde{O}(\epsilon m)$ functions (the soft-O notation suppresses dependence on quantities beside $\epsilon$, $m$ and $\beta$, as well as logarithmic terms) are not $L$-Lipschitz for any pair of points at distance $\epsilon$ in the domain $\mathbb{C}$. That is, for all $m$ and for all $\varepsilon \geq m^{-\beta}$,*

$$\mathbb{E}\left[\max_{\substack{\rho, \rho' \in \mathbb{C} \\ \|\rho - \rho'\|_2 \leq \epsilon}} \left\|\left|\{i \in [m] \mid \ell_i(\rho) - \ell_i(\rho') > L\|\rho - \rho'\|_2\}\right|\right\|\right] \leq \tilde{O}(\epsilon m) \tag{3}$$

Assuming a sequence of $m$ $\beta$-dispersed loss functions and initial distribution $w_1$ set to the uniform distribution over $C$ and optimize the step size parameter, the exponential forecaster presented in Algorithm 1 achieves sublinear regret $\tilde{O}(\sqrt{dm\log(Rm)} + (L+1)m^{1-\beta})$. While this result achieves a no-regret procedure, its lack of dependence on both the task-data and on the chosen initialization makes it difficult to meta-learn. In the following theorem, we generalize the regret bound for the exponential forecaster to make it data-dependent and hyperparameter-dependent:

**Theorem 2.1.** *Let $\ell_1, \dots, \ell_m : C \mapsto [0,1]$ be any sequence of piecewise L-Lipschitz functions that are $\beta$-dispersed. Suppose $C \subset \mathbb{R}^d$ is contained in a ball of radius R. The exponentially weighted forecaster (Algorithm 1) has expected regret $\mathbf{R}_m \leq m\lambda + \frac{\log(1/Z)}{\lambda} + \tilde{O}((L+1)m^{1-\beta})$, where $Z = \frac{\int_{\mathcal{B}(\rho^*, m^{-\beta})} w(\rho)d\rho}{\int_C w(\rho)d\rho}$ for $\rho^*$ the optimal action in hindsight.*

The proof of this result adapts past analyses of Algorithm 1; setting step-size $\lambda$ appropriately recovers the previously mentioned bound. The new bound is useful due to its explicit dependence on both the initialization $w$ and the optimum in hindsight via the $\log(1/Z)$ term. Assuming $w$ is a (normalized) distribution, this effectively measures the overlap between the chosen initialization and a small ball around the optimum; we thus call $-\log Z = -\log \frac{\int_{\mathcal{B}(\rho^*, m^{-\beta})} w(\rho)d\rho}{\int_C w(\rho)d\rho}$ the **negative log-overlap** of initialization $w(.)$ with the optimum $\rho^*$.

We also obtain an asymptotic lower bound on the expected regret of any algorithm by extending the argument of [10] to the multi-task setting. We show that for finite $D^*$ we must suffer $\tilde{\Omega}(m^{1-\beta})$ regret, which limits the improvement we can hope to achieve from task-similarity.

**Theorem 2.2.** *There is a sequence of piecewise L-Lipschitz $\beta$-dispersed functions $\ell_{i,j} : [0,1] \mapsto [0,1]$, whose optimal actions in hindsight $\arg\min_\rho \sum_{i=1}^m l_{t,i}(\rho)$ are contained in some fixed ball of diameter $D^*$, for which any algorithm has expected regret $\mathbf{R}_m \geq \tilde{\Omega}(m^{1-\beta})$.*

## 2.1 Task-similarity

Before proceeding to our discussion of meta-learning, we first discuss what we might hope to achieve with it; specifically, we consider what a reasonable notion of task-similarity is in this setting. Note that the Theorem 2.1 regret bound has three terms, of which two depend on the hyperparameters and the last is due to dispersion and cannot be improved via better settings. Our focus will thus be on improving the first two terms, which are the dominant ones due to the dependence on the dimensionality and the distance from the initialization encoded in the negative log overlap. In particular, when the initialization is the uniform distribution then this quantity depends inversely on the size of a small ball around the optimum, which may be quite small. Via meta-learning we hope to assign more of the probability mass of the initializer to areas close to the optimum, which will decrease these terms. On average, rather than a dependence on the volume of a small ball we aim to achieve a dependence on the **average negative log-overlap**

$$V^2 = -\min_{w: C \mapsto \mathbb{R}_{\geq 0}, \int_C w(\rho)d\rho = 1} \frac{1}{T} \sum_{t=1}^T \log \int_{\mathcal{B}(\rho_t^*, m^{-\beta})} w(\rho)d\rho \tag{4}$$

which can be much smaller if the task optima $\rho_t^*$ are close together; for example, if they are the same then $V = 0$, corresponding to assigning all the initial weight within the common ball $\mathcal{B}(\rho^*, m^{-\beta})$ around the shared optima. This is also true if $\text{vol}(\cap_{t \in T} \mathcal{B}(\rho_t^*, m^{-\beta})) > 0$, as one can potentially initialize with all the weight in the intersection of the balls. On the other hand if $\text{vol}(\cap_{t \in T} \mathcal{B}(\rho_t^*, m^{-\beta})) = 0$, $V > 0$. For example, if a $p$-fraction of tasks have optima $\rho_0$ and the remaining at $\rho_1$ with $||\rho_0 - \rho_1|| > 2m^{-\beta}$ the task similarity is given by the binary entropy function $V = H_b(p) = -p\log p - (1-p)\log(1-p)$.

The settings of Algorithm 1 that achieve the minimum in the definition of $V$ are directly related to $V$ itself: the optimal initializer is the distribution achieving $V$ and the optimal step-size is $V/\sqrt{m}$. Note that while the explicit definition requires computing a minimum over a set of functions, the task-similarity can be computed using the discretization constructed in Section 3.1.

# 3 An algorithm for meta-learning the initialization and step-size

Having established a single-task algorithm and shown how its regret depends on the initialization and step-size, we move on to meta-learning these hyperparameters. Recall that our goal is to make the task-averaged regret (2) small, in particular to improve upon the baseline of repeatedly running Algorithm 1 from the uniform distribution, up to $o_T(1)$ terms that vanish as we see more tasks. This accomplishes the meta-learning goal of using multiple tasks to improve upon single-task learning.

In this paper, we use the strategy of running online learning algorithms on the data-dependent regret guarantees from above [29]. If we can do so with sublinear regret in $T$, then we will improve upon the single-task guarantees up to $o_T(1)$ terms, as desired. Specifically, we are faced with a sequence of regret-upper-bounds $U_t(w, v) = (v + f_t(w)/v)\sqrt{m} + g(m)$ on nonnegative functions $w$ over $C$ and positive scalars $v > 0$. Note that $g(m)$ cannot be improved via meta-learning, so we will focus on learning $w$ and $v$. To do so, we run two online algorithms, one over the functions $f_t$ and the other over $h_t(v) = v + f_t(w_t)/v$, where $w_t$ is set by the first procedure. As shown in the following result, if both procedures have sublinear regret then our task-averaged regret will have the desired properties:

**Theorem 3.1.** *Assume each task $t \in [T]$ consists of a sequence of $m$ $\beta$-dispersed piecewise $L$-Lipschitz functions $\ell_{t,i} : C \mapsto [0, 1]$. Let $f_t$ and $g$ be functions such that the regret of Algorithm 1 run with step-size $\lambda = v\sqrt{m}$ for $v > 0$ and initialization $w : C \mapsto \mathbb{R}_{\geq 0}$ is bounded by $U_t(w, v) = (v + f_t(w)/v)\sqrt{m} + g(m)$. Suppose we have a procedure that achieves $F_T(w)$ regret w.r.t. any $w : C \mapsto \mathbb{R}_{\geq 0}$ by playing actions $w_t : C \mapsto \mathbb{R}_{\geq 0}$ on $f_t$ and another procedure that achieves $H_T(v)$ regret w.r.t. any $v > 0$ by playing actions $v_t > 0$ on $h_t(v) = v + f_t(w_t)/v$, where $H_T$ is non-increasing on the positive reals. Then by setting $\rho_{t,i}$ using Algorithm 1 with step-size $v_t/\sqrt{m}$ and initialization $w_t$ at each task $t$, for $w^* = \arg\min_{w:C\mapsto\mathbb{R}_{\geq 0}} \sum_{t=1}^T f_t(w)$ the optimal initialization and $V$ the task-similarity (4) we get task-averaged regret bounded by*

$$\left(\frac{H_T(V)}{T} + \min\left\{\frac{F_T(w^*)}{VT}, 2\sqrt{F_T(w^*)/T}\right\} + 2V\right)\sqrt{m} + g(m) \qquad (5)$$

This result is an analog of [29, Theorem 3.1] and follows by manipulating the definition of regret. It reduces the problem of obtaining a small task-averaged regret to solving two online learning problems, one to set the initialization and one to set the step-size. So long as both have sublinear regret then we will improve over single-task learning. In the next two sections we derive suitable procedures.

## 3.1 Meta-learning the initialization

The most technically challenging component of the procedure is learning the initialization. As discussed, this can be done via a no regret procedure for the sequence $f_t(w) = -\log\frac{\int_{\mathcal{B}(\rho_t^*, m^{-\beta})} w(\rho)d\rho}{\int_C w(\rho)d\rho}$. This is nontrivial as the optimization domain is a set of nonnegative functions or measures on the domain $C$. To handle this, we introduce some notations and abstractions. At each task $t$ we face a function $f_t$ associated with an unknown closed subset $C_t \subset C$—in particular $C_t = \mathcal{B}(\rho_t^*, m^{-\beta})$—with positive volume $\text{vol}(C_t) > 0$ that is revealed after choosing $w_t : C \mapsto \mathbb{R}_{\geq 0}$ For each time $t$ define the discretization $\mathcal{D}_t = \{D = \bigcap_{s \leq t} C_s^{(\mathbf{c}_{[s]})} : \mathbf{c} \in \{0, 1\}^t, \text{vol}(D) > 0\}$ of $C$, where $C_t^{(0)} = C_t$ and $C_t^{(1)} = C\backslash C_t$. We will use elements of these discretizations to index nonnegative vectors in $\mathbb{R}_{\geq 0}^{|\mathcal{D}_t|}$; specifically, for any measure $w : C \mapsto \mathbb{R}_{\geq 0}$ let $\mathbf{w}(t) \in \mathbb{R}_{\geq 0}^{|\mathcal{D}_t|}$ denote the vector with entries $\mathbf{w}(t)_{[D]} = \int_D w(\rho)d\rho$ for $D \in \mathcal{D}_t$. Note that we will exclusively use $p, q, v, w$ for measures, with $v$ specifically referring to the uniform measure, i.e. $\mathbf{v}(t)_{[D]} = \text{vol}(D)$. For convenience, for all real vectors $\mathbf{x}$ we will use $\hat{\mathbf{x}}$ to denote $\mathbf{p}/\|\mathbf{p}\|_1$. Finally, we abuse notation and remove the parentheses to refer those vectors associated with the final discretization, i.e. $\mathbf{v} = \mathbf{v}(T)$ and $\mathbf{w} = \mathbf{w}(T)$.

Now that we have this notation we can turn back to the functions we are interested in: $f_t(w) = -\log\frac{\int_{C_t} w(\rho)d\rho}{\int_C w(\rho)d\rho}$, where $C_t = \mathcal{B}(\rho_t^*, m^{-\beta})$. Observe that we can equivalently write this as $f_t(\mathbf{w}) = -\log\langle \mathbf{w}_t^*, \hat{\mathbf{w}}\rangle$, where $\mathbf{w}_{t[D]}^* = 1_{D \subset C_t}$; this translates our online learning problem from the domain of measures on $C$ to the simplex on $|\mathcal{D}_T|$ elements. However, we cannot play in this domain explicitly as we do not have access to the final discretization $\mathcal{D}_T$, nor do we get access to $\mathbf{w}_t^*$ after task $t$, except implicitly via $C_t$. In this section we design a method that implicitly run an online convex optimization procedure over $\mathbb{R}_{\geq 0}^{|\mathcal{D}_T|}$ while explicitly playing probability measures $w : C \mapsto \mathbb{R}_{\geq 0}$.

---
**Algorithm 2** Follow-the-Regularized-Leader (prescient form)
---
1: **Input:** discretization $\mathcal{D}_T$ of $C$, mixture parameter $\gamma \in [0, 1]$, step-size $\eta > 0$
2: Initialize $\mathbf{w}_1 = \hat{\mathbf{v}}$
3: **for** $t = 1, 2, \ldots, T$ **do**
4:   Play $\mathbf{w}_t$ and suffer $f_t(\mathbf{w}_t) = -\log\langle \mathbf{w}_t^*, \mathbf{w}_t\rangle$.
5:   Observe $f_t$.
6:   Update $\mathbf{w}_{t+1} = \arg\min_{\|\mathbf{w}\|_1=1, \mathbf{w}\geq\gamma\hat{\mathbf{v}}} D_{KL}(\mathbf{w}||\hat{\mathbf{v}}) + \eta \sum_{s\leq t} f_s(\mathbf{w})$
---

As the functions $f_t$ are exp-concave, one might first consider applying a method attaining logarithmic regret on such losses [28, 37]; however, such algorithms have regret that depends linearly on the dimension, which in our case is poly($T$). We thus turn to the the follow-the-regularized-leader (FTRL) family of algorithms, which in the case of entropic regularization are well-known to have regret logarithmic in the dimension [43]. In Algorithm 2 we display the pseudo-code of a modification with regularizer $D_{KL}(\cdot||\hat{\mathbf{v}})$, where recall $\mathbf{v}$ is the vector of volumes of the discretization $\mathcal{D}_T$ of $C$, and we constrain the played distribution to have measure at least $\gamma\hat{\mathbf{v}}_{[D]}$ over every set $D \in \mathcal{D}_T$. While this requires knowing the discretization $\mathcal{D}_T$ of $C$ in advance, the following lemma shows that we can run the procedure knowing only the discretization $\mathcal{D}_t$ after task $t$ by simply minimizing the same objective over distributions discretized on $\mathcal{D}_t$. This crucially depends on the re-scaling of the entropic regularizer by $\hat{\mathbf{v}}$ (i.e. the uniform distribution over $C$) and the fact that $\mathbf{w}_t^* \in \{0, 1\}^{|\mathcal{D}_T|}$.

**Lemma 3.1.** *Let $w : C \mapsto \mathbb{R}_{\geq 0}$ be the probability measure corresponding to the minimizer*

$$\mathbf{w} = \arg\min_{\|\mathbf{q}\|_1=1, \mathbf{q}\geq\gamma\hat{\mathbf{v}}} D_{KL}(\mathbf{q}||\hat{\mathbf{v}}) - \eta \sum_{s\leq t} \log\langle \mathbf{w}_s^*, \mathbf{q}\rangle \tag{6}$$

*and let $\tilde{w} : C \mapsto \mathbb{R}_{\geq 0}$ be the probability measure corresponding to the minimizer*

$$\tilde{\mathbf{w}}(t) = \arg\min_{\|\mathbf{q}\|_1=1, \mathbf{q}\geq\gamma\hat{\mathbf{v}}(t)} D_{KL}(\mathbf{q}||\hat{\mathbf{v}}(t)) - \eta \sum_{s\leq t} \log\langle \mathbf{w}_s^*(t), \mathbf{q}\rangle \tag{7}$$

*Then $\mathbf{w} = \tilde{\mathbf{w}}$.*

We can thus move on to proving a regret guarantee for Algorithm 2. This follows from Jensen's inequality together with standard results for FTRL once we show that the loss functions are $\frac{1}{\gamma \mathrm{vol}(C_t)}$-Lipschitz over the constrained domain, yielding the following guarantee for Algorithm 2:

**Theorem 3.2.** *Algorithm 2 has regret w.r.t. $\mathbf{w}^* \in \arg\min_{\|\mathbf{w}\|_1=1, \mathbf{w}\geq\mathbf{0}} \sum_{t=1}^{T} f_t(\mathbf{w})$ bounded by*

$$\frac{1-\gamma}{\eta} D_{KL}(\mathbf{w}^*||\hat{\mathbf{v}}) + \frac{\eta}{\gamma^2} \sum_{t=1}^{T} \frac{1}{(\mathrm{vol}(C_t))^2} + \gamma \sum_{t=1}^{T} \log \frac{1}{\mathrm{vol}(C_t)} \tag{8}$$

*Setting $\gamma^2 = GB/\sqrt{T}$ and $\eta^2 = \frac{B^2\gamma^2}{TG^2}$, where $B^2 = D_{KL}(\mathbf{w}^*||\hat{\mathbf{v}})$ and $G^2 = \frac{1}{T}\sum_{t=1}^{T} \frac{1}{(\mathrm{vol}(C_t))^2}$, yields sublinear regret $\tilde{O}(\sqrt{BG}T^{\frac{3}{4}})$.*

*Proof.* Algorithm 2 is standard FTRL with regularizer $\frac{1}{\eta} D_{KL}(\cdot||\hat{\mathbf{v}})$, which has the same Hessian as the standard entropic regularizer over the simplex and is thus $\frac{1}{\eta}$-strongly-convex w.r.t. $\|\cdot\|_1$ [43, Example 2.5]. Applying Jensen's inequality, the standard regret bound for FTRL [43, Theorem 2.11] together with the Lipschitz guarantee of Claim B.1, and Jensen's inequality again yields the result:

$$\sum_{t=1}^{T} f_t(\mathbf{w}_t) - f_t(\mathbf{w}^*) = \sum_{t=1}^{T} f_t(\mathbf{w}_t) - (1-\gamma)f_t(\mathbf{w}^*) - \gamma f_t(\hat{\mathbf{v}}) + \gamma(f_t(\hat{\mathbf{v}}) - f_t(\mathbf{w}^*))$$

$$\leq \sum_{t=1}^{T} f_t(\mathbf{w}_t) - f_t(\gamma\hat{\mathbf{v}} + (1-\gamma)\mathbf{w}^*) + \gamma \log \frac{\langle \mathbf{w}_t^*, \mathbf{w}^*\rangle}{\langle \mathbf{w}_t^*, \hat{\mathbf{v}}\rangle}$$

$$\leq \frac{1}{\eta} D_{KL}(\gamma\hat{\mathbf{v}} + (1-\gamma)\mathbf{w}^*||\hat{\mathbf{v}}) + \frac{\eta}{\gamma^2} \sum_{t=1}^{T} \frac{1}{(\mathrm{vol}(C_t))^2} + \gamma \sum_{t=1}^{T} \log \frac{1}{\mathrm{vol}(C_t)}$$

$$\leq \frac{1-\gamma}{\eta} D_{KL}(\mathbf{w}^*||\hat{\mathbf{v}}) + \frac{\eta}{\gamma^2} \sum_{t=1}^{T} \frac{1}{(\mathrm{vol}(C_t))^2} + \gamma \sum_{t=1}^{T} \log \frac{1}{\mathrm{vol}(C_t)}$$

$\square$

Since the regret is sublinear in $T$, this result satisfies our requirement for attaining asymptotic improvement over single-task learning via Theorem 3.1. However, there are several aspects of this bound that warrant some discussion. The first is the rate of $T^{\frac{3}{4}}$, which is less sublinear than the standard $\sqrt{T}$ and certainly the $\log T$ regret of exp-concave functions. However, the functions we face are (a) non-Lipschitz and (b) over a domain that has dimensionality $\Omega(T)$; both violate conditions for good rates in online convex optimization [28, 43], making our problem much more difficult.

A more salient aspect is the dependence on $B^2 = D_{KL}(\mathbf{w}^*||\hat{\mathbf{v}})$, effectively the negative entropy of the optimal initialization. This quantity is in-principle unbounded but is analogous to standard online convex optimization bounds that depend on the norm of the optimum, which in e.g. the Euclidean case are also unbounded. In our case, if the optimal distribution is highly concentrated on a very small subset of the space it will be difficult to compete with. Note that our setting of $\eta$ depends on knowing or guessing $B$; this is also standard but is certainly a target for future work to address. For example, past work on parameter-free algorithms has solutions for optimization over the simplex [38]; however, it is unclear whether this is straightforward to do while preserving the property given by Lemma 3.1 allowing us to implicitly work with an unknown discretization. A more reasonable approach may be to compete only with smooth measures that only assign probability at most $\kappa \operatorname{vol}(D)$ to any subset $D \subset C$ for some constant $\kappa \geq 1$; in this case we will simply have $B$ bounded by $\log \kappa$.

A final issue is the dependence on $\sqrt{G}$, which is bounded by the reciprocal of the smallest volume $\operatorname{vol}(C_t)$, which in the dispersed case is roughly $O(m^{\beta d})$; this means that the task-averaged regret will have a term that, while decreasing as we see additional tasks, is *increasing* in the number of within-task iterations and the dispersion parameter, which is counter-intuitive. It is also does so exponentially in the dimension. Note that in the common algorithm configuration setting of $\beta = 1/2$ and $d = 1$ this will simply mean that for each task we suffer an extra $o_T(1)$ loss at each within-task round, a quantity which vanishes asymptotically.

## 3.2 Meta-learning the step-size

In addition to learning the initialization, Theorem 3.1 requires learning the task-similarity to set the within-task step-size $\lambda > 0$. This involves optimizing functions of form $h_t(v) = v + f_t(w_t)/v$. Since we know that the measures $w_t$ are lower-bounded in terms of $\gamma$, we can apply a previous result [29] that solves this by running the EWOO algorithm [28] on the modified sequence $v + \frac{f_t(w_t)+\varepsilon^2}{v}$:

**Corollary 3.1.** *For any $\varepsilon > 0$, running the EWOO algorithm on the modified sequence $v + \frac{f_t(w)+\varepsilon^2}{v}$ over the domain $[\varepsilon, \sqrt{D^2 - \log \gamma + \varepsilon^2}]$, where $D^2 \geq \frac{1}{T}\sum_{t=1}^{T} \log \frac{1}{\operatorname{vol}(C_t)}$, attains regret*

$$\min\left\{\frac{\varepsilon^2}{v^*}, \varepsilon\right\} T + \frac{\sqrt{D^2 - \log \gamma}}{2} \max\left\{\frac{D^2 - \log \gamma}{\varepsilon^2}, 1\right\}(1 + \log(T+1)) \qquad (9)$$

*on the original sequence $h_t(v) = v + f_t(w)/v$ for all $v^* > 0$.*

Setting $\varepsilon = 1/\sqrt[4]{T}$ gives a guarantee of form $\tilde{O}((\min\{1/v^*, \sqrt[4]{T}\})\sqrt{T})$. Note this rate might be improvable by using the fact that $v$ is lower-bounded due to the $\gamma$-constraint; however, we do not focus on this since this component is not the dominant term in the regret. In fact, because of this we can adapt a related method that simply runs follow-the-leader (FTL) on the same modified sequence [29] without affecting the dominant terms in the regret:

**Corollary 3.2.** *For any $\varepsilon > 0$, running the FTL algorithm on the modified sequence $v + \frac{f_t(w)+\varepsilon^2}{v}$ over the domain $[\varepsilon, \sqrt{D^2 - \log \gamma + \varepsilon^2}]$, where $D^2 \geq \frac{1}{T}\sum_{t=1}^{T} \log \frac{1}{\operatorname{vol}(C_t)}$, attains regret*

$$\min\left\{\frac{\varepsilon^2}{v^*}, \varepsilon\right\} T + 2\sqrt{D^2 - \log \gamma} \max\left\{\frac{(D^2 - \log \gamma)^{\frac{3}{2}}}{\varepsilon^3}, 1\right\}(1 + \log(T+1)) \qquad (10)$$

*on the original sequence $h_t(v) = v + f_t(w)/v$ for all $v^* > 0$.*

Setting $\varepsilon = 1/\sqrt[5]{T}$ gives a guarantee of form $\tilde{O}((\min\{1/v^*, \sqrt[5]{T}\})T^{\frac{3}{5}})$. The alternatives are described in pseudocode at the bottom of Algorithm 3; while the guarantee of the FTL-based approach is worse, it is almost as simple to compute as the task-similarity and does not require integration, making it easier to implement.

---

**Algorithm 3** Meta-learning the parameters of the exponential forecaster (Algorithm 1). Recall that $\mathbf{p}(t)$ refers to the time-$t$ discretization of the measure $p : C \mapsto \mathbb{R}_{\geq 0}$ (c.f. Section 3.1).

---

1: **Input:** domain $C \subset \mathbb{R}^d$, dispersion $\beta > 0$, step-size $\eta > 0$, constraint parameter $\gamma \in [0, 1]$, offset parameter $\varepsilon > 0$, domain parameter $D > 0$.

2: Initialize $w_1$ to the uniform measure on $C$ and set $\lambda_1 = \frac{\varepsilon + \sqrt{D^2 + \varepsilon^2 - \log \gamma}}{2\sqrt{m}}$.

3: **for** task $t = 1, 2, \ldots, T$ **do**

4:     Run Algorithm 1 with initialization $w_t$ and step-size $\lambda_t$ and obtain task-$t$ optimum $\rho_t^* \in C$.

5:     Set $w_t^* = 1_{\mathcal{B}(\rho_t^*, m^{-\beta})}$ to be the function that is 1 in the $m^{-\beta}$-ball round $\rho_t^*$ and 0 elsewhere.

6:     Set $w_{t+1}$ to $\mathbf{w}_{t+1}(t) = \arg\min_{\|\mathbf{w}\|_1 = 1, \mathbf{w} \geq \gamma \hat{\mathbf{v}}(t)} D_{KL}(\mathbf{w} || \hat{\mathbf{v}}(t)) - \eta \sum_{s \leq t} \log \langle \mathbf{w}_s^*(t), \mathbf{w} \rangle$.

7:     **if** using EWOO **then**

8:         Define $\mu_t(x) = \exp\left(-\alpha\left(tx + \frac{t\varepsilon^2 - \sum_{s \leq t} \log \langle \mathbf{w}_s^*(s), \mathbf{w}_s(s) \rangle}{x}\right)\right)$ for $\alpha = \frac{2}{D} \min\left\{\frac{\varepsilon^2}{D^2}, 1\right\}$.

9:         Set $\lambda_{t+1} = \frac{\int_\varepsilon^{\sqrt{D^2 + \varepsilon^2 - \log \gamma}} x \mu_t(x) dx}{\sqrt{m} \int_\varepsilon^{\sqrt{D^2 + \varepsilon^2 - \log \gamma}} \mu_t(x) dx}$.

10:     **else**

11:         Set $\lambda_{t+1} = \sqrt{\frac{\sum_{s \leq t} \varepsilon^2 - \log \langle \mathbf{w}_s^*(s), \mathbf{w}_s(s) \rangle}{tm}}$.

---

## 3.3 Putting the two together

We now combine our methods for the initialization and step-size in Algorithm 3 to meta-learn the parameters of the exponential forecaster. This yields a task-averaged regret bound via Theorem 3.1:

**Theorem 3.3.** *Define* $B^2 = D_{KL}(\mathbf{w}^* || \hat{\mathbf{v}})$, $G^2 = \frac{1}{T} \sum_{t=1}^T \frac{1}{(\text{vol}(C_t))^2}$, *and* $D^2 \geq \frac{1}{T} \sum_{t=1}^T \log \frac{1}{\text{vol}(C_t)} = O(\beta d \log m)$. *Then Algorithm 3 with* $\eta, \gamma$ *set as in Theorem 3.2 and* $\varepsilon = 1/\sqrt[4]{T}$ *(if using EWOO) or* $1/\sqrt[5]{T}$ *(otherwise) yields task-averaged regret*

$$\tilde{O}\left(\min\left\{\frac{\sqrt{BG}}{V\sqrt[4]{T}}, \frac{\sqrt[4]{BG}}{\sqrt[8]{T}}\right\} + 2V\right)\sqrt{m} + g(m) \tag{11}$$

As in past meta-learning work this achieves the goal of adapting to the task-similarity, attaining asymptotic average regret of $2V\sqrt{m} + O(m^{-\beta})$, where we substitute for the dispersion term $g$ and $V^2$ is the task-similarity encoding the average probability mass assigned to different task balls by the optimal initialization. We include the minimum of two rates in the bound: $1/\sqrt[4]{T}$ if the task-similarity is a constant $\Theta_T(1)$ and $1/\sqrt[8]{T}$ if it is extremely small. As discussed, this reflects the difficulty of our meta-problem, in which we are optimizing non-smooth functions over distributions; in contrast, past meta procedures take advantage of nice properties of Bregman divergences to obtain faster rates [29].

# 4 Meta-learning for data-driven algorithm design

We demonstrate the utility of our bounds in a series of applications across data-driven algorithm design and robust learning. This section focuses on the former and demonstrates how our results imply guarantees for meta-learning the tuning of solvers for difficult combinatorial problems. We also demonstrate the practical utility of our approach for tuning clustering on real and synthetic datasets.

## 4.1 Instantiations for tuning combinatorial optimization algorithms

Algorithm configuration for combinatorial optimization algorithms involves learning algorithm parameters from multiple instances of combinatorial problems [27, 12, 4]. For well-known problems like MWIS (maximum weighted independent set), IQP (integer quadratic programming) and mechanism design for auctions, the algorithmic performance on a fixed instance is typically a piecewise Lipschitz function of the algorithm parameters. Prior work has looked at learning these parameters in the distributional setting (i.e. assuming iid draws of problem instances) [12] or the online setting where the problem instances may be adversarially drawn [11, 10]. On the other hand, instantiating our results for these problems provide upper bounds for much more realistic settings where different tasks may be related and our bounds improve with this relatedness.

We demonstrate how to apply our results to combinatorial problems under mild smoothness assumptions. The key idea is to show that if inputs come from a smooth distribution, the performance is dispersed (as a sequence of functions in the algorithm parameters). We demonstrate this for the greedy algorithm for the knapsack problem and for initialization in $k$-center clustering. Similar results may be obtained for other problems, e.g. MWIS, IQP, and auction design (c.f. Appendix C).

**Greedy Knapsack.** Knapsack is a well-known NP-complete problem. We are given a knapsack with capacity cap and items $i \in [m]$ with sizes $w_i$ and values $v_i$. The goal is to select a subset $S$ of items to add to the knapsack such that $\sum_{i \in S} w_i \leq$ cap while maximizing the total value $\sum_{i \in S} v_i$ of selected items. The greedy heuristic to add items in decreasing order of $v_i/w_i$ gives a 2-approximation. We consider a generalization to use $v_i/w_i^\rho$ proposed by [27] for $\rho \in [0, 10]$. For example, for the value-weight pairs $\{(0.99, 1), (0.99, 1), (1.01, 1.01)\}$ and capacity cap $= 2$ the classic heuristic $\rho = 1$ gives value 1.01 but using $\rho = 3$ gives the optimal value 1.98. We can learn the optimal $\rho$ from similar tasks, and obtain the following formal guarantees (proof in Appendix C).

**Theorem 4.1.** *Consider instances of the knapsack problem given by bounded weights $w_{i,j} \in [1, C]$ and $\kappa$-bounded independent values $v_{i,j} \in [0, 1]$ for $i \in [m], j \in [T]$. Then the task-averaged regret for learning the parameter $\rho$ for the greedy heuristic family above is $o_T(1) + 2V\sqrt{m} + O(\sqrt{m})$.*

**$k$-center clustering.** We consider the $\alpha$-Llyod's algorithm family introduced in [15]. In the seeding phase, each point $x$ is sampled with probability $\propto \min_{c \in C} d(v, c)^\alpha$, where $d(\cdot, \cdot)$ is the distance metric and $C$ is the set of centers chosen so far. The family contains an algorithm for each $\alpha \in [0, \infty) \cup \infty$, and includes popular clustering heuristics like randomly initialized $k$-means ($\alpha = 0$), $k$-means++ ($\alpha = 2$) and farthest-first traversal ($\alpha = \infty$). Performance is measured using the Hamming distance to the best clustering and is a piecewise constant function of $\alpha$. Our result can be instantiated for this problem even without smoothness by leveraging the randomness of the clustering algorithm.

**Theorem 4.2.** *Consider instances of the $k$-center clustering problem on $n$ points, with Hamming loss $l_{i,j}$ for $i \in [m], j \in [T]$ against some (unknown) ground truth. Then the task-averaged regret for learning the parameter $\alpha$ for the $\alpha$-Lloyd's algorithm family [15] is $o_T(1) + 2V\sqrt{m} + O(\sqrt{m})$.*

## 4.2 Experiments for greedy knapsack and $k$-center clustering

Next we demonstrate our meta-initialization algorithm empirically on knapsack and $k$-center clustering. We design experiments on real and simulated data that show the usefulness of our techniques in learning a sequence of piecewise-Lipschitz functions. For knapsack, we generate a synthetic dataset of instances as follows. For each problem instance of each task, we have cap $= 100$ and $m = 50$. We have 10 'heavy' items with $w_i \sim \mathcal{N}(27, 0.5)$ and $v_i \sim \mathcal{N}(27, 0.5)$, and 40 items with $w_i \sim \mathcal{N}(19 + w_t, 0.5)$ and $v_i \sim \mathcal{N}(18, 0.5)$, where $w_t \in [0, 2]$ is task-dependent.

We also consider the $\alpha$-Lloyd's algorithm family introduced in [15]. The performance of the algorithm is measured using the Hamming loss relative to the best clustering and is a piecewise constant function of $\alpha$. We can compute the pieces for $\alpha \in [0, 10]$ by iteratively computing the subset of parameter values where a candidate point can be the next center. We use the small split of the *Omniglot* dataset [33], and create clustering tasks by drawing random samples consisting of five characters each, where four characters are constant throughout. We also create a Gaussian mixture binary classification dataset where each class is a 2D diagonal Gaussian distribution consisting of 100 points each with variance $\sigma$ and $2\sigma$ and centers $(0, 0)$ and $(d\sigma, 0)$. We pick $d \in [2, 3]$ to create different tasks.

We learn using 30 instances each of 10 training tasks and evaluate average loss over 5 test tasks. We use 100 trials to average over the randomization of the clustering algorithm and the exponential forecaster algorithm. We perform meta-initialization with parameters $\gamma = \eta = 0.01$ (no hyperparameter search performed). The step-size is set to minimize the regret term in Theorem 2.1, and not meta-learned.

The relative improvement in task-averaged regret due to meta-learning in our formal guarantees depend on the task-similarity $V$ and how it compares to the dispersion-related $O(m^{1-\beta})$ term, and can be significant when the latter is small. Our results in Table 1 show that meta-learning an initialization, i.e. a distribution over the algorithm parameter, for the exponential forecaster in this setting yields improved performance on each dataset. We observe this for both the one-shot and five-shot settings, i.e. the number of within-task iterations of the test task are one and five, respectively. The benefit of meta-learning is most pronounced for the Gaussian mixture (well-dispersed and similar tasks), and gains for Omniglot may increase with more tasks (dispersed but less similar tasks). For knapsack, the relative gains are smaller (similar tasks, but less dispersed). See Appendix D for further experiments.

Table 1: Effect of meta-initialization on few-shot learning of algorithmic parameters. Performance is as a fraction of the average value (Hamming accuracy, or knapsack value) of the offline optimum.

| Dataset | Omniglot | | Gaussian Mixture | | Knapsack | |
|---|---|---|---|---|---|---|
| | One-shot | Five-shot | One-shot | Five-shot | One-shot | Five-shot |
| Single task | $88.67 \pm 0.47\%$ | $95.02 \pm 0.19\%$ | $90.10 \pm 1.10\%$ | $91.43 \pm 0.44\%$ | $84.74 \pm 0.29\%$ | $98.89 \pm 0.17\%$ |
| Meta-initialized | $89.65 \pm 0.49\%$ | $96.05 \pm 0.15\%$ | $95.76 \pm 0.60\%$ | $96.39 \pm 0.27\%$ | $85.66 \pm 0.57\%$ | $99.12 \pm 0.15\%$ |

## 5 Robust online meta-learning

In online learning, we aim to minimize a sequence of losses and want to perform well relative to the optimum in hindsight. It is possible for the observed losses to be noisy on some inputs, either naturally or due to an adversary. We explore the conditions under which robustness to adversarial influence (i.e. outlier injection) is possible, which is common in meta-learning with diverse sources.

*Setup*: At round $i$, we play $x_i$, observe perturbed loss $\tilde{l}_i : \mathcal{X} \to [0,1]$ which is set by the adversary by modifying the true loss $l_i : \mathcal{X} \to [0,1]$ using an *attack function* $a_i : \mathcal{X} \to [0,1]$ such that $\tilde{l}_i = l_i + a_i$ and may be non-Lipschitz, and suffer perturbed loss $\tilde{l}_i(x_i)$ and true loss $l_i(x_i)$. We seek to minimize regret relative to best fixed action in hindsight, i.e. $\tilde{R}_m = \sum_{i=1}^{m} \tilde{l}_i(x_i) - \min_{x \in \mathcal{X}} \sum_{i=1}^{m} \tilde{l}_i(x)$ for the perturbed loss and regret $R_m = \sum_{i=1}^{m} l_i(x_i) - \min_{x \in \mathcal{X}} \sum_{i=1}^{m} l_i(x)$ for the true loss.

No regret can be achieved provided the adversary distribution is sufficiently smooth, i.e. satisfies $\beta$-dispersion for some $\beta > 0$, as this corresponds to online optimization of the perturbed loss function. We can show this for both perturbed and true loss. The perturbed loss guarantee is immediate from standard results on online learning of piecewise Lipschitz functions [11, 10]. For the true loss, we can achieve no regret if the adversary perturbation $a_i$ is limited to small balls and the centers of the balls are dispersed, which we capture using the following definition.

**Definition 5.1** ($\delta$-bounded, $\beta_a$-dispersed attack). *An attack function $a_i$ is $\delta$-bounded if there exists a ball $\mathcal{B}(x_a, \delta)$ of radius $\delta$ such that $a_i(x) = 0$ for each $x \in \mathcal{X} \setminus \mathcal{B}(x_a, \delta)$. $x_a$ is called a center $c_{a_i}$ for attack $a_i$. A sequence of attack functions $a_1, \ldots, a_m$ is said to be $\beta_a$-dispersed, if the positions of attack centers $x_a$ are dispersed i.e. for all $m$ and for all $\epsilon \geq m^{-\beta_a}$, $\mathbb{E}\left[\max_{x,x' \in \mathcal{X}, x \in \mathcal{B}(x', \epsilon)} \left|\{i \in [m] \mid x = c_{a_i}\}\right|\right] \leq \tilde{O}(\epsilon m)$.*

**Theorem 5.1.** *Given a sequence of $\beta$-dispersed adversarially perturbed losses $\tilde{l}_i = l_i + a_i$, where $\tilde{l}_i, l_i, a_i$ are piecewise $L$-Lipschitz functions $\mathcal{X} \to [0,1]$ for $i = 1, \ldots, m$ and $\mathcal{X} \subset \mathbb{R}^d$, the exponential forecaster algorithm has $\mathbb{E}[\tilde{R}_m] = \tilde{O}(m\lambda + \frac{\log(1/Z)}{\lambda} + (L+1)m^{1-\beta})$ (with $Z$ as in Theorem 2.1). If in addition we have that $a_i$ is a $m^{-\beta_a}$-bounded, $\beta_a$-dispersed attack, then $\mathbb{E}[R_m] = \tilde{O}(m\lambda + \frac{\log(1/Z)}{\lambda} + (L+1)m^{1-\min\{\beta, \beta_a\}})$.*

Together with Theorem 3.3, this implies no regret meta-learning in the presence of dispersed adversaries, in particular the occurrence of unreliable data in small dispersed parts of the domain. We also show a lower bound which establishes that our bounds are essentially optimal in the attack dispersion.

**Theorem 5.2.** *There exist sequences of piecewise $L$-Lipschitz functions $\tilde{l}_i, l_i, a_i$ $[0,1] \to [0,1]$ for $i = 1, \ldots, m$ such that for any online algorithm*

1. *$\tilde{l}_i$ is $\beta$-dispersed and $\mathbb{E}[\tilde{R}_m] = \Omega(m^{1-\beta})$,*

2. *$\tilde{l}_i$ is $\beta$-dispersed, $a_i$ is $m^{-\beta}$-bounded, $\beta_a$-dispersed and $\mathbb{E}[R_m] = \Omega(m^{1-\min\{\beta, \beta_a\}})$.*

## 6 Conclusion

In this paper we studied the initialization-based meta-learning of piecewise-Lipschitz functions, demonstrating how online convex optimization over an adaptive discretization can find an initialization that improves the performance of the exponential forecaster across tasks, assuming the tasks have related optima. We then applied this result in two settings: online configuration of clustering algorithms and adversarial robustness in online learning. For the latter we introduced a dispersion-based understanding of robustness that we believe to be of independent interest. In addition, there are further interesting applications of our work to other algorithm configuration problems.

## Acknowledgments

This material is based on work supported by the National Science Foundation under grants CCF-1535967, CCF-1910321, IIS-1618714, IIS-1901403, SES-1919453, IIS-1705121, IIS-1838017, IIS-2046613 and IIS-2112471; the Defense Advanced Research Projects Agency under cooperative agreements HR00112020003 and FA875017C0141; an AWS Machine Learning Research Award; an Amazon Research Award; a Bloomberg Research Grant; a Microsoft Research Faculty Fellowship; an Amazon Web Services Award; a Facebook Faculty Research Award; funding from Booz Allen Hamilton Inc.; a Block Center Grant; and a Two Sigma Fellowship Award. Any opinions, findings and conclusions or recommendations expressed in this material are those of the author(s) and do not necessarily reflect the views of any of these funding agencies.

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
