## A   Related work

The success of meta-learning has led to significant theoretical effort to understand it. Most efforts studying initialized-based meta-learning focus on the convex Lipschitz setting [20, 30]; work studying inherently nonconvex modeling approaches instead usually study multi-task representation learning [6, 35, 22, 47] or target optimization, e.g. stationary point convergence [24]. An exception is a study of linear models over Gaussian data showing that nonconvexity is critical to meta-learning an initialization that exploits low-rank task structure [42]. There is also work extending results from the neural tangent kernel literature to meta-learning [49], but in this case the objective becomes convex. On the other hand, we study initializations for learning a class of functions that can be highly non-convex and have numerous discontinuities. Theoretically, our work uses the Average Regret-Upper-Bound Analysis (ARUBA) strategy [29] for obtaining a meta-update procedure for initializing within-task algorithms, which has been applied elsewhere for privacy [34] and federated learning [31]; the main technical advance in our work is in providing the guarantees for it in our setting, which is challenging due to the need to learn over a space of probability measures. Another aspect of our work is to learn the step-size of the within-task algorithm, in addition to its initialization; our approach is similar to that of ARUBA [29] but the step-size alone has also been tuned in several works in a variety of different settings [27, 31, 48].

Data-driven configuration is the selection of an algorithm from a parameterized family, by doing learning over multiple problem instances [27, 12]. In other words, it is 'hyperparameter tuning' with formal guarantees, and has applications to integer programming, clustering and learning with limited labeled data [9, 7, 14]. In this work, we show how this general approach can be made even more effective by enabling it to adapt to task similarity. We also show applications of our results to robust meta-learning in the presence of outliers in the dataset [39, 32]. While previous work on robust online learning has considered adversaries with bounded perturbation in the online learning setting [1, 41], our results allow potentially unbounded perturbations provided the adversary uses a smooth distribution. That is, the adversarial attack can be thought of as a distribution of perturbations, similar to the smoothed analysis approach of [44]. In the offline setting, a similar attack is studied in the context of deep network feature-space attacks by [5]. We also remark that our formulation has a poisoning aspect, since we do not observe the clean loss $l_h(x)$, which is of particular interest in federated learning [3, 46]. Also, note that unlike the typical applications of data-driven design where optimization is over the dual loss function, i.e. loss as a function of the algorithm parameter for a fixed sample $x \in \mathcal{X}$, here we consider learning loss or confidence functions over the input space $\mathcal{X}$.

More background on data-driven algorithm selection, an algorithm design paradigm for setting algorithm parameters when multiple instances of a problem are available or need to be solved, can be found in [16, 4]. By modeling the problem of identifying a good algorithm from data as a statistical learning problem, general learning algorithms have been developed which exploit smoothness of the underlying algorithmic distribution [11]. This provides a new algorithmic perspective, along with tools and insights for good performance under this smoothed analysis for fundamental problems including clustering, mechanism design, and mixed integer programs, and providing guarantees like differential privacy, adaptive online learning and adversarial robustness [7, 13, 10, 5].

## B   Proofs

### B.1   Proof of Theorem 2.1

*Proof.* The proof adapts the analysis of the exponential forecaster in [11]. Let $W_t = \int_C w_t(\rho)d\rho$ be the normalizing constant and $P_t = \mathbb{E}_{\rho \sim p_t}[u_t(\rho)]$ be the expected payoff at round $t$. Also let $U_t(\rho) = \sum_{j=1}^{t} u_j(\rho)$. We seek to bound $R_T = OPT - P(T)$, where $OPT = U_T(\rho^*)$ for optimal parameter $\rho^*$ and $P(T) = \sum_{t=1}^{T} P_t$ is the expected utility of Algorithm 1 in $T$ rounds. We will do this by lower bounding $P(T)$ and upper bounding $OPT$ by analyzing the normalizing constant $W_t$.

*Lower bound for $P(T)$*: This follows from standard arguments, included for completeness. Using the definitions in Algorithm 1, it follows that

$$\frac{W_{t+1}}{W_t} = \frac{\int_{\mathbb{C}} e^{\lambda u_t(\rho)} w_t(\rho)d\rho}{W_t} = \int_{\mathbb{C}} e^{\lambda u_t(\rho)} \frac{w_t(\rho)}{W_t}d\rho = \int_{\mathbb{C}} e^{\lambda u_t(\rho)} p_t(\rho)d\rho.$$

Use inequalities $e^{\lambda x} \leq 1 + (e^\lambda - 1)x$ for $x \in [0,1]$ and $1 + x \leq e^x$ to conclude

$$\frac{W_{t+1}}{W_t} \leq \int_{\mathbb{C}} p_t(\rho) \left(1 + (e^\lambda - 1)u_t(\rho)\right) d\rho = 1 + (e^{H\lambda} - 1)P_t \leq \exp\left((e^\lambda - 1)P_t\right).$$

Finally, we can write $W_{T+1}/W_1$ as a telescoping product to obtain

$$\frac{W_{T+1}}{W_1} = \prod_{t=1}^{T} \frac{W_{t+1}}{W_t} \leq \exp\left((e^\lambda - 1)\sum_t P_t\right) = \exp\left(P(T)(e^\lambda - 1)\right),$$

or, $W_{T+1} \leq \exp\left(P(T)(e^\lambda - 1)\right) \int_C w_1(\rho)d\rho$.

*Upper bound for* $OPT$: Let $\mathcal{B}^*(r)$ be the ball of radius $r$ around $\rho^*$. If there are at most $k$ discontinuities in any ball of radius $r$, we can conclude that for all $\rho \in \mathcal{B}^*(r)$, $U_T(\rho) \geq OPT - k - LTr$. Now, since $W_{T+1} = \int_C w_1(\rho) \exp(\lambda U_T(\rho))d\rho$, we have

$$\begin{aligned}
W_{T+1} &\geq \int_{\mathcal{B}^*(r)} w_1(\rho)e^{\lambda U_T(\rho)}d\rho \\
&\geq \int_{\mathcal{B}^*(r)} w_1(\rho)e^{\lambda(OPT-k-LTr)}d\rho \\
&= e^{\lambda(OPT-k-LTr)} \int_{\mathcal{B}^*(r)} w_1(\rho)d\rho.
\end{aligned}$$

Putting together with the lower bound, and rearranging, gives

$$\begin{aligned}
OPT - P_T &\leq \frac{P(T)(e^\lambda - 1 - \lambda)}{\lambda} + \frac{\log(1/Z)}{\lambda} + k + LTr \\
&\leq T\lambda + \frac{\log(1/Z)}{\lambda} + k + LTr,
\end{aligned}$$

where we use that $P(T) \leq T$ and for all $x \in [0,1], e^x \leq 1 + x + (e-2)x^2$. Take expectation over the sequence of utility functions and apply dispersion to conclude the result. $\qquad\square$

## B.2 Proof of Theorem 2.2

We extend the construction in [10] to the multi-task setting. The main difference is that we generalize the construction for any task similarity, and show that we get the same lower bound asymptotically.

*Proof.* Define $u^{(b,x)}(\rho) = I[b = 0] * I[\rho > x] + I[b = 1] * I[\rho \leq x]$, where $b \in \{0,1\}$, $x, \rho \in [0,1]$ and $I[\cdot]$ is the indicator function. For each iteration the adversary picks $u^{(0,x)}$ or $u^{(1,x)}$ with equal probability for some $x \in [a, a + D^*]$, the ball of diameter $D^*$ containing all the optima.

For each task $t$, $m - \frac{3}{D^*}m^{1-\beta}$ functions are presented with the discontinuity $x \in [a + D^*/3, a + 2D^*/3]$ while ensuring $\beta$-dispersion. The remaining $\frac{3}{D^*}m^{1-\beta}$ are presented with discontinuities located in successively halved intervals (the 'halving adversary') containing the optima in hindsight, any algorithm gets half of these wrong in expectation. It is readily verified that the functions are $\beta$-dispersed. The construction works provided $m$ is sufficiently large ($m > \left(\frac{3}{D^*}\right)^{1/\beta}$). The task averaged regret is therefore also $\tilde{\Omega}(m^{1-\beta})$. $\qquad\square$

### B.3 Proof of Theorem 3.1

*Proof.*

$$\sum_{t=1}^{T}\sum_{m=1}^{m}\ell_{t,i}(\rho_{t,i}) - \min_{\rho_t^* \in C}\sum_{i=1}^{m}\ell_{t,i}(\rho_t^*)$$

$$\leq \sum_{t=1}^{T}U_t(w_t, v_t)$$

$$\leq \min_{v>0}H_T(v)\sqrt{m} + \sum_{t=1}^{T}\left(v + \frac{f_t(w_t)}{v}\right)\sqrt{m} + g(m)$$

$$\leq \min_{w:C\mapsto\mathbb{R}_{\geq 0},v>0}H_T(v)\sqrt{m} + \frac{F_T(w)\sqrt{m}}{v} + \sum_{t=1}^{T}\left(v + \frac{f_t(w)}{v}\right)\sqrt{m} + g(m)$$

$$\leq \left(H_T(V) + \min\left\{\frac{F_T(w^*)}{V}, 2\sqrt{F_T(w^*)T}\right\} + 2TV\right)\sqrt{m} + Tg(m)$$

where the last step is achieved by substituting $w = w^*$ and $v = \max\left\{V, \sqrt{F_T(w^*)/T}\right\}$. $\qquad\square$

### B.4 Proof of Lemma 3.1

*Proof.* Define a probability measure $p : C \mapsto \mathbb{R}_{\geq 0}$ that is constant on all elements $\tilde{D} \in \mathcal{D}_t$ of the discretization at time $t$, taking the value $p(\rho) = \frac{1}{\text{vol}(\tilde{D})}\sum_{D\in\mathcal{D}_T, D\subset\tilde{D}}\mathbf{w}_{[D]} \;\forall\, \rho \in \tilde{D}$. Note that for any $D \in \mathcal{D}_T$ that is a subset of $\tilde{D}$ we have that

$$\mathbf{p}_{[D]} = \int_D \tilde{w}(\rho)d\rho = \frac{\mathbf{V}_{[D]}}{\sum_{D'\in\mathcal{D}_T, D'\subset\tilde{D}}\mathbf{V}_{[D']}}\sum_{D'\in\mathcal{D}_T, D'\subset\tilde{D}}\mathbf{w}_{[D']}$$

Then

$$D_{KL}(\mathbf{p}||\hat{\mathbf{v}}) - \eta\sum_{s\leq t}\log\langle\mathbf{w}_s^*, \mathbf{p}\rangle$$

$$= \sum_{\tilde{D}\in\mathcal{D}_t}\sum_{D\in\mathcal{D}_T, D\subset\tilde{D}}\mathbf{p}_{[D]}\log\frac{\mathbf{p}_{[D]}}{\hat{\mathbf{v}}_{[D]}} - \eta\sum_{s\leq t}\log\sum_{\tilde{D}\in\mathcal{D}_t}\sum_{D\in\mathcal{D}_T, D\subset\tilde{D}}\mathbf{w}_{s[D]}^*\mathbf{p}_{[D]}$$

$$= \sum_{\tilde{D}\in\mathcal{D}_t}\sum_{D\in\mathcal{D}_T, D\subset\tilde{D}}\frac{\mathbf{V}_{[D]}}{\sum_{D'\in\mathcal{D}_T, D'\subset\tilde{D}}\mathbf{V}_{[D']}}\sum_{D'\in\mathcal{D}_T, D'\subset\tilde{D}}\mathbf{w}_{[D']}\log\frac{\sum_{D'\in\mathcal{D}_T, D'\subset\tilde{D}}\mathbf{w}_{[D']}}{\sum_{D'\in\mathcal{D}_T, D'\subset\tilde{D}}\hat{\mathbf{v}}_{[D']}}$$

$$- \eta\sum_{s\leq t}\log\sum_{\tilde{D}\in\mathcal{D}_t}\sum_{D\in\mathcal{D}_T, D\subset\tilde{D}}\frac{\mathbf{w}_{s[D]}^*\mathbf{V}_{[D]}}{\sum_{D'\in\mathcal{D}_T, D'\subset\tilde{D}}\mathbf{V}_{[D']}}\sum_{D'\in\mathcal{D}_T, D'\subset\tilde{D}}\mathbf{w}_{[D']}$$

$$\leq \sum_{\tilde{D}\in\mathcal{D}_t}\sum_{D\in\mathcal{D}_T, D\subset\tilde{D}}\frac{\mathbf{V}_{[D]}}{\sum_{D'\in\mathcal{D}_T, D'\in\tilde{D}}\mathbf{V}_{[D']}}\sum_{D'\in\mathcal{D}_T, D'\subset\tilde{D}}\mathbf{w}_{D'}\log\frac{\mathbf{w}_{[D']}}{\hat{\mathbf{v}}_{[D']}}$$

$$- \eta\sum_{s\leq t}\log\sum_{\tilde{D}\in\mathcal{D}_t, \tilde{D}\subset C_s}\sum_{D\in\mathcal{D}_T, D\subset\tilde{D}}\frac{\mathbf{V}_{[D]}}{\sum_{D'\in\mathcal{D}_T, D'\subset\tilde{D}}\mathbf{V}_{[D']}}\sum_{D'\in\mathcal{D}_T, D'\subset\tilde{D}}\mathbf{w}_{[D']}$$

$$= \sum_{\tilde{D}\in\mathcal{D}_t}\sum_{D'\in\mathcal{D}_T, D'\in\tilde{D}}\mathbf{w}_{[D']}\log\frac{\mathbf{w}_{[D']}}{\hat{\mathbf{v}}_{[D']}} - \eta\sum_{s\leq t}\log\sum_{\tilde{D}\in\mathcal{D}_t, \tilde{D}\subset C_s}\sum_{D'\in\mathcal{D}_T, D'\subset\tilde{D}}\mathbf{w}_{[D']}$$

$$= D_{KL}(\mathbf{w}||\hat{\mathbf{v}}) - \eta\sum_{s\leq t}\log\langle\mathbf{w}_s^*, \mathbf{w}\rangle$$

where the inequality follows from applying the log-sum inequality to the first term and the fact that $\mathbf{w}_{s[D]}^* = \mathbf{1}_{D\subset C_s}$ in the second term. Note that we also have

$$\|\mathbf{p}\|_1 = \sum_{\tilde{D}\in\mathcal{D}_t}\sum_{D\in\mathcal{D}_T, D\subset\tilde{D}}\frac{\mathbf{V}_{[D]}}{\sum_{D'\in\mathcal{D}_T, D'\subset\tilde{D}}\mathbf{V}_{[D']}}\sum_{D'\in\mathcal{D}_T, D'\subset\tilde{D}}\mathbf{w}_{[D']} = \sum_{\tilde{D}\in\mathcal{D}_t}\sum_{D'\in\mathcal{D}_T, D'\subset\tilde{D}}\mathbf{w}_{[D']} = 1$$

and

$$\mathbf{P}_{[D]} = \frac{\mathbf{v}_{[D]}}{\sum_{D'\in\mathcal{D}_T,D'\subset\tilde{D}}\mathbf{v}_{[D']}}\sum_{D'\in\mathcal{D}_T,D'\subset\tilde{D}}\mathbf{w}_{[D']} \geq \frac{\gamma\mathbf{v}_{[D]}}{\sum_{D'\in\mathcal{D}_T,D'\subset\tilde{D}}\mathbf{v}_{[D']}}\sum_{D'\in\mathcal{D}_T,D'\subset\tilde{D}}\hat{\mathbf{v}}_{[D']} = \gamma\hat{\mathbf{v}}_{[D]}$$

so $\mathbf{p}$ satisfies the optimization constraints. Therefore, since $\mathbf{w}$ was defined to be the minimum of the sum of the KL-divergence (a strongly-convex function [43, Example 2.5]) and a convex function, it is unique and so coincides with $\mathbf{p}$.

On the other hand

$$D_{KL}(\mathbf{p}(t)||\hat{\mathbf{v}}(t)) - \eta\sum_{s\leq t}\log\langle\mathbf{w}_s^*(t),\mathbf{p}(t)\rangle \leq D_{KL}(\mathbf{p}||\hat{\mathbf{v}}) - \eta\sum_{s\leq t}\log\langle\mathbf{w}_s^*,\mathbf{p}\rangle$$

$$= D_{KL}(\mathbf{w}||\hat{\mathbf{v}}) - \eta\sum_{s\leq t}\log\langle\mathbf{w}_s^*,\mathbf{w}\rangle$$

$$\leq D_{KL}(\tilde{\mathbf{w}}||\hat{\mathbf{v}}) - \eta\sum_{s\leq t}\log\langle\mathbf{w}_s^*,\tilde{\mathbf{w}}\rangle$$

$$= D_{KL}(\tilde{\mathbf{w}}(t)||\hat{\mathbf{v}}(t)) - \eta\sum_{s\leq t}\log\langle\mathbf{w}_s^*(t),\tilde{\mathbf{w}}(t)\rangle$$

where the first inequality follows from above and the second from the optimality of $\mathbf{w}$. Note that by nonnegativity the discretization of $p$ does not affect its measure over $C$, so $\|\mathbf{p}\|_1 = 1 \implies \|\mathbf{p}(t)\|_1 = 1$. Finally, also from above we have

$$\mathbf{p}(t)_{[D]} = \sum_{D'\in\mathcal{D}_T,D'\subset D}\mathbf{p}_{[D']} \geq \gamma\sum_{D'\in\mathcal{D}_T,D'\subset D}\mathbf{p}_{[D']}\hat{\mathbf{v}}_{[D']} = \gamma\hat{\mathbf{v}}(t)_{[D]}$$

Thus as before $\mathbf{p}(t)$ satisfies the optimization constraints, which with the previous inequality and the uniqueness of the optimum $\tilde{\mathbf{w}}(t)$ implies that $\mathbf{p}(t) = \tilde{\mathbf{w}}(t)$. Finally, since $\tilde{w}$ is constant on all elements of the discretization $\mathcal{D}_t$ of $C$ this last fact implies that $\mathbf{p} = \tilde{\mathbf{w}}$, which together with $\mathbf{p} = \mathbf{w}$ implies the result. $\square$

## B.5 Lipschitzness for Algorithm 2

**Claim B.1.** *The loss $f_t$ is $\frac{1}{\gamma\operatorname{vol}(C_t)}$-Lipschitz w.r.t. $\|\cdot\|_1$ over the set $\{\mathbf{w}\in\mathbb{R}^{|\mathcal{D}_T|} : \|\mathbf{w}\|_1 = 1, \mathbf{w}\geq \gamma\hat{\mathbf{v}}\}$.*

*Proof.*

$$\max_{\|\mathbf{w}\|_1=1,\mathbf{w}\geq\gamma\hat{\mathbf{v}}}\|\nabla\log\langle\mathbf{w}_t^*,\mathbf{w}\rangle\|_\infty = \max_{D,\|\mathbf{w}\|_1=1,\mathbf{w}\geq\gamma\hat{\mathbf{v}}}\frac{\mathbf{w}_t^*{}_{[D]}}{\langle\mathbf{w}_t^*,\mathbf{w}\rangle} \leq \frac{1}{\langle\mathbf{w}_t^*,\gamma\hat{\mathbf{v}}\rangle} = \frac{1}{\gamma\operatorname{vol}(C_t)}$$

$\square$

## B.6 Proof of Corollary 3.1

*Proof.* Using first-order conditions we have that the optimum in hindsight of the functions $h_t$ satisfies

$$v^2 = \frac{1}{T}\sum_{t=1}^T f_t(w_t) = -\frac{1}{T}\sum_{t=1}^T\log\langle\mathbf{w}_t^*,\mathbf{w}_t\rangle \leq \frac{1}{T}\sum_{t=1}^T\log\frac{1}{\gamma\operatorname{vol}(C_t)}$$

Applying [29, Corollary C.2] with $\alpha_t = 1$, $B_t^2 = f_t(w_t)$, and $D^2 - \log\gamma$ instead of $D^2$ yields the result. $\square$

## B.7 Proof of Corollary 3.2

*Proof.* Using first-order conditions we have that the optimum in hindsight of the functions $h_t$ satisfies

$$v^2 = \frac{1}{T}\sum_{t=1}^T f_t(w_t) = -\frac{1}{T}\sum_{t=1}^T\log\langle\mathbf{w}_t^*,\mathbf{w}_t\rangle \leq \frac{1}{T}\sum_{t=1}^T\log\frac{1}{\gamma\operatorname{vol}(C_t)}$$

Applying [29, Proposition B.2] with $\alpha_t = 1$, $B_t^2 = f_t(w_t)$, and $D^2 - \log\gamma$ instead of $D^2$ yields the result. $\square$

### B.8 Proof of Theorem 3.3

*Proof.* We have $F_T(w^*) = \tilde{O}(\sqrt{BG}T^{\frac{3}{4}})$ and $H_T(V) = \tilde{O}(\min\{1/V, \sqrt[5]{T}\}T^{\frac{3}{5}})$ from Corollaries 3.1 and 3.2. Substituting into Lemma 3.1 and simplifying yields

$$\tilde{O}\left(\frac{\min\left\{\frac{1}{V}, \sqrt[4]{T}\right\}}{\sqrt{T}} + \min\left\{\frac{\sqrt{BG}}{V\sqrt[4]{T}}, \frac{\sqrt[4]{BG}}{\sqrt[8]{T}}\right\} + 2V\right)\sqrt{m} + g(m)$$

Simplifying further yields the result. $\qquad\square$

### B.9 Proof of Theorem 5.1

*Proof.* The bound on $\mathbb{E}[\tilde{R}_T]$ is immediate from Theorem 2.1. For $\mathbb{E}[R_T]$, we can upper bound the natural regret with the sum of robust regret, total adversarial perturbation at the optimum and a term corresponding to the difference between the loss of natural and robust optima.

$$R_T = \sum_{t=1}^{T} l_t(x_t) - \min_{x\in\mathcal{X}}\sum_{t=1}^{T} l_t(x)$$

$$= \tilde{R}_T + \sum_{t=1}^{T} l_t(x_t) - \sum_{t=1}^{T} \tilde{l}_t(x_t) + \min_{x\in\mathcal{X}}\sum_{t=1}^{T} \tilde{l}_t(x) - \min_{x\in\mathcal{X}}\sum_{t=1}^{T} l_t(x)$$

$$= \tilde{R}_T - \sum_{t=1}^{T} a_t(x_t) + \sum_{t=1}^{T} a_t(\tilde{x}^*) + \sum_{t=1}^{T} l_t(\tilde{x}^*) - \sum_{t=1}^{T} l_t(x^*)$$

$$\leq \tilde{R}_T + \sum_{t=1}^{T} a_t(\tilde{x}^*) + \left|\sum_{t=1}^{T} l_t(\tilde{x}^*) - \sum_{t=1}^{T} l_t(x^*)\right|$$

where $\tilde{x}^* = \arg\min_{x\in\mathcal{X}}\sum_{t=1}^{T}\tilde{l}_t(x)$ and $x^* = \arg\min_{x\in\mathcal{X}}\sum_{t=1}^{T} l_t(x)$. We now use the $\beta_a$-dispersedness of the attack to show an excess expected regret of $\tilde{O}(T^{1-\beta_a})$. Using attack dispersion on a ball of radius $T^{-\beta_a}$ around $\tilde{x}^*$, the number of attacks that have non-zero $a_t(\tilde{x}^*)$ is at most $\tilde{O}(T^{1-\beta_a})$, and therefore $\sum_{t=1}^{T} a_t(\tilde{x}^*) \leq \tilde{O}(T^{1-\beta_a})$. Further, observe that the robust and natural optima coincide unless some attack occurs at the natural optimum $x^*$. We can use attack dispersion at $x^*$, and a union bound across rounds, to conclude $\mathbb{E}|\sum_{t=1}^{T} l_t(\tilde{x}^*) - \sum_{t=1}^{T} l_t(x^*)| \leq \tilde{O}(T^{1-\beta_a})$ which concludes the proof. $\qquad\square$

### B.10 Proof of Theorem 5.2

*Proof.* Part 1 follows from the lower bound in Theorem 2.2, by setting $\tilde{l}_i = l_i$ as the loss sequence used in the proof.

To establish Part 2, we extend the construction as follows. $\tilde{l}_i = l_i$ are both equal and correspond to the 'halving adversary' from the proof of Theorem 2.2 for the first $\Theta(m^{1-\beta})$ rounds. If $\beta \leq \beta_a$ we are done, so assume otherwise. Let $I$ denote the interval containing the optima over the rounds so far. Notice that the length of $I$ is at most $|I| \leq (\frac{1}{2})^{\Theta(m^{1-\beta})} \leq (\frac{1}{2})^{\beta\log m} = m^{-\beta}$ for $\beta > 0$. For further rounds $l_i$ continues to be the halving adversary for $\Theta(m^{1-\beta_a})$ rounds, which implies any algorithm suffers $\Omega(m^{1-\beta_a})$ regret. We set attack $a_i$ on interval $I$ such that $\tilde{l}_i = 0$ on $I$ on these rounds. This ensures that $a_i$ is $\beta_a$-dispersed and $\tilde{l}_i$ is $\beta$-dispersed. Putting together with the case $\beta \leq \beta_a$, we obtain $\Omega(m^{1-\min\{\beta, \beta_a\}})$ bound on the regret of any algorithm. $\qquad\square$

## C  Learning algorithmic parameters for combinatorial problems

We discuss implications of our results for several combinatorial problems of widespread interest including integer quadratic programming and auction mechanism design. We will need the following theorem from [14], which generalizes the recipe for establishing dispersion given by [8] for $d = 1, 2$ dimensions to arbitrary constant $d$ dimendions. It is straightforward to apply the recipe to establish

dispersion for these problems, which in turn implies that our meta-learning results are applicable. We demonstrate this for a few important problems below for completeness.

**Theorem C.1** ([14]). *Let $l_1, \ldots, l_m : \mathbb{R}^d \to \mathbb{R}$ be independent piecewise L-Lipschitz functions, each having discontinuities specified by a collection of at most $K$ algebraic hypersurfaces of bounded degree. Let $\mathcal{L}$ denote the set of axis-aligned paths between pairs of points in $\mathbb{R}^d$, and for each $s \in \mathcal{L}$ define $D(m, s) = |\{1 \le t \le m \mid l_t \text{ has a discontinuity along } s\}|$. Then we have $\mathbb{E}[\sup_{s \in \mathcal{L}} D(m, s)] \le \sup_{s \in \mathcal{L}} \mathbb{E}[D(m, s)] + O(\sqrt{m \log(mK)})$.*

## C.1 Greedy knapsack

We are given a knapsack with capacity cap and items $i \in [m]$ with sizes $w_i$ and values $v_i$. The goal is to select a subset $S$ of items to add to the knapsack such that $\sum_{i \in S} w_i \le \text{cap}$ while maximizing the total value $\sum_{i \in S} v_i$ of selected items. We consider a general greedy heuristic to insert items with largest $v_i / w_i^\rho$ first (due to [27]) for $\rho \in [0, 10]$.

The classic greedy heuristic sets $\rho = 1$ and can be used to provide a 2-approximation for the problem. However other values of $\rho$ can improve the knapsack objective on certain problem instances. For example, for the value-weight pairs $\{(0.99, 1), (0.99, 1), (1.01, 1.01)\}$ and capacity cap $= 2$ the classic heuristic $\rho = 1$ gives value 1.01 as the greedy heuristic is maximized for the third item. However, using $\rho = 3$ (or any $\rho > 1 + \log(1/0.99)/\log(1.01) > 2.01$) allows us to pack the two smaller items giving the optimal value 1.98.

Our result (Theorem 3.3) when applied to this problem shows that it is possible to learn the optimal parameter values for the greedy heuristic algorithm family for knapsack from similar tasks.

**Theorem C.2.** *Consider instances of the knapsack problem given by bounded weights $w_{i,j} \in [1, C]$ and $\kappa$-bounded independent values $v_{i,j} \in [0, 1]$ for $i \in [m], j \in [T]$. Then the asymptotic task-averaged regret for learning the algorithm parameter $\rho$ for the greedy heuristic family described above is $o_T(1) + 2V\sqrt{m} + O(\sqrt{m})$.*

*Proof.* Lemma 11 of [8] shows that the loss functions form a $\frac{1}{2}$-dispersed sequence. The result follows by applying Theorem 3.3 with $\beta = \frac{1}{2}$. $\square$

## C.2 $k$-center clustering

We consider the $\alpha$-Lloyd's clustering algorithm family from [15], where the initial $k$ centers in the procedure are set by sampling points with probability proportional to $d^\alpha$ where $d$ is the distance from the centers selected so far for some $\alpha \in [0, D], D \in \mathbb{R}_{\ge 0}$. For example, $\alpha = 0$ corresponds to the vanilla $k$-means with random initial centers, and $\alpha = 2$ setting is the $k$-means++ procedure. For this algorithm family, we are able to show the following guarantee. Interestingly, for this family it is sufficient to rely on the internal randomness of the algorithmic procedure and we do not need assumptions on data smoothness.

**Theorem C.3.** *Consider instances of the $k$-center clustering problem on $n$ points, with Hamming loss $l_{i,j}$ for $i \in [m], j \in [T]$ against some (unknown) ground truth clustering. Then the asymptotic task-averaged regret for learning the algorithm parameter $\alpha$ for the $\alpha$-Lloyd's clustering algorithm family of [15] is $o_T(1) + 2V\sqrt{m} + O(\sqrt{m})$.*

*Proof.* We start by applying Theorem 4 from [15] to an arbitrary $\alpha$-interval $[\alpha_0, \alpha_0 + \epsilon] \subseteq [0, D]$ of length $\epsilon$. The expected number of discontinuities (expectation under the internal randomness of the algorithm when sampling successive centers), is at most

$$D(m, \epsilon) = O(nk \log(n) \log(\max\{(\alpha_0 + \epsilon)/\alpha_0), (\alpha_0 + \epsilon) \log R\}),$$

where $R$ is an upper bound on the ratio between any pair of non-zero distances. Considering cases $\alpha_0 \lessgtr \frac{1}{\log R}$ and using the inequality $\log(1 + x) \le x$ for $x \ge 0$ we get that there are, in expectation, at most $O(\epsilon nk \log n \log R)$ discontinuities in any interval of length $\epsilon$. Theorem C.1 now implies $\frac{1}{2}$-dispersion using the recipe from [8]. The task-averaged regret bound follows from Theorem 3.3. $\square$

## C.3 Maximum weighted independent set (MWIS)

We leverage known results about the MWIS problem to show $\frac{1}{2}$-dispersion, which together with Theorem 3.3 implies that our bound on the task-averaged regret improves with task similarity $V$.

In MWIS, there is a graph $G = (V, E)$ and a weight $w_v \in \mathbb{R}^+$ for each vertex $v \in V$. The goal is to find a set of non-adjacent vertices with maximum total weight. The problem is $NP$-hard and in fact does not have any constant factor polynomial time approximation algorithm. [27] propose a greedy heuristic family, which selects vertices greedily based on largest value of $w_v/(1 + \deg(v))^\rho$, where $\deg(v)$ is the degree of vertex $v$, and removes neighbors of the selected vertex before selecting the next vertex.

For this algorithm family, we can learn the best parameter $\rho$ provided pairs of vertex weights have a joint $\kappa$-bounded distribution, and Theorem 3.3 implies regret bounds that improve with task similarity. We use the recipe from [8] to establish dispersion.

**Theorem C.4.** *Consider instances of MWIS with all vertex weights in* $(0, 1]$ *and for each instance, every pair of vertex weights has a $\kappa$-bounded joint distribution. Then the asymptotic task-averaged regret for learning the algorithm parameter $\rho$ is* $o_T(1) + 2V\sqrt{m} + O(\sqrt{m})$.

*Proof.* The loss function is piecewise constant with discontinuities corresponding to $\rho$ such that $w_v/(1 + \deg(v))^\rho = w_u/(1 + \deg(u))^\rho$ for a pair of vertices $u, v$. [11] show that the discontinuities have $(\kappa \ln n)$-bounded distributions where $n$ is the number of vertices. This implies that in any interval of length $\epsilon$, we have in expectation at most $\epsilon \kappa \ln n$ discontinuities. Using this in dispersion recipe from [8] implies $\frac{1}{2}$-dispersion, which in turn implies the desired regret bound by applying Theorem 3.3. $\qquad\square$

## C.4 Integer quadratic programming (IQP)

The objective is to maximize a quadratic function $z^T A z$ for $A$ with non-negative diagonal entries, subject to $z \in \{0, 1\}^n$. In the classic Goemans-Williamson algorithm [26] one solves an SDP relaxation $U^T A U$ where columns $u_i$ of $U$ are unit vectors. $u_i$ are then rounded to $\{\pm 1\}$ by projecting on a vector $Z$ drawn according to the standard Gaussian, and using $\texttt{sgn}(\langle u_i, Z \rangle)$. A simple parametric family is $s$-linear rounding where the rounding is as before if $|\langle u_i, Z \rangle| > s$ but uses probabilistic rounding to round $u_i$ to 1 with probability $\frac{1 + (\langle u_i, Z \rangle)/s}{2}$. The dispersion analysis of the problem from [11] and the general recipe from [8] imply that our results yield low task-averaged regret for learning the parameter of the $s$-linear rounding algorithms.

**Theorem C.5.** *Consider instances of IQP given by matrices $A_{i,j}$ and rounding vectors $Z_{i,j} \sim \mathcal{N}_n$ for $i \in [m], j \in [T]$. Then the asymptotic task-averaged regret for learning the algorithm parameter $s$ for $s$-linear rounding is* $o_T(1) + 2V\sqrt{m} + O(\sqrt{m})$.

*Proof.* As noted in [11], since $Z_{i,j}$ are normal, the local of discontinuities $s = |\langle u_i, Z \rangle|$ are distributed with a $\sqrt{\frac{2}{\pi}}$-bounded density. Thus in any interval of length $\epsilon$, we have in expectation at most $\epsilon \sqrt{\frac{2}{\pi}}$ discontinuities. Theorem C.1 together with the general recipe from [8] implies $\frac{1}{2}$-dispersion. The task-averaged regret bound is now a simple application of Theorem 3.3. $\qquad\square$

Our results are an improvement over prior work which have only considered iid and (single-task) online learning settings. Similar improvements can be obtained for auction design, as described below. We illustrate this using a relatively simple auction, but the same idea applies for an extensive classes of auctions as studied in [13].

## C.5 Posted price mechanisms with additive valuations

There are $m$ items and $n$ bidders with valuations $v_j(b_i), j \in [n], i \in [2^m]$ for all $2^m$ bundles of items. We consider additive valuations which satisfy $v_j(b) = \sum_{i \in b} v_j(\{i\})$. The objective is to maximize the social welfare (sum of buyer valuations). If the item values for each buyer have $\kappa$-bounded distributions, then the corresponding social welfare is dispersed and our results apply.

**Theorem C.6.** *Consider instances of posted price mechanism design problems with additive buyers and $\kappa$-bounded marginals of item valuations. Then the asymptotic task-averaged regret for learning the price which maximizes the social welfare is $o_T(1) + 2V\sqrt{m} + O(\sqrt{m})$.*

*Proof.* As noted in [11], the locations of discontinuities are along axis-parallel hyperplanes (buyer $j$ will be willing to buy item $i$ at a price $p_i$ if and only if $v_j(\{i\}) \geq p_i$, each buyer-item pair in each instance corresponds to a hyperplane). Thus in any pair of points $p, p'$ (corresponding to pricing) at distance $\epsilon$, we have in expectation at most $\epsilon\kappa mn$ discontinuities along any axis-aligned path joining $p, p'$, since discontinuities for an item can only occur along axis-aligned segment for the axis corresponding to the item. Theorem C.1 now implies $\frac{1}{2}$-dispersion. The task-averaged regret bound is now a simple application of Theorem 3.3. $\qquad\square$

## D  Additional experiments

We include additional experiments to study how the meta-learning improves with number of tasks, and how the variance of the performance of our approach across different datasets can be understood by examining task similarity and data dispersion.

### D.1  Number of training tasks needed for meta-learning

We also examine the number of training tasks that our meta-learning procedure needs to obtain improvements over the single-task baseline. We use a single test task, and a variable number of training tasks (0 through 10) to meta-learn the initialization. We use the same settings as in Section 4.2, except the meta-learning experiments have been averaged over 20 iterations (to average over randomization in the algorithms). In Figure 1, we plot the average regret against number of meta-updates performed before starting the test task, and compare against the single-task baselines. We observe gains with meta-learning with just $T = 10$ tasks for the Omniglot dataset, and with even a single task in the Gaussian mixture dataset. The latter is likely due to a very high degree of task similarity across all the tasks (examined below), so learning on any task transfers very well to another task.

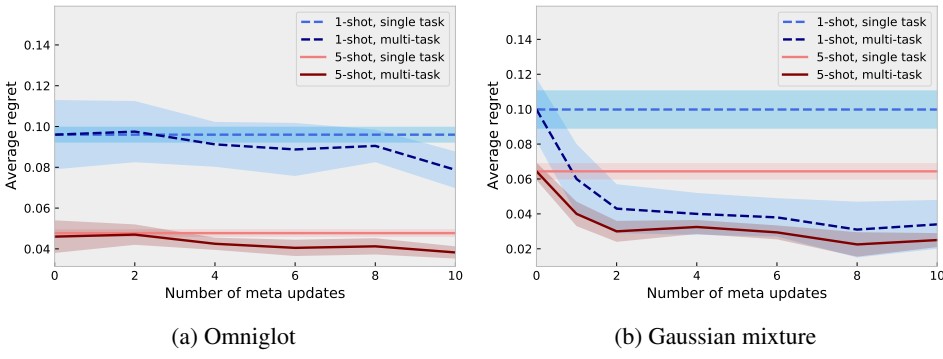

(a) Omniglot                    (b) Gaussian mixture

Figure 1: Average regret vs. number of training tasks for meta-learning.

### D.2  Task similarity and dispersion

We also examine the task similarity of the different tasks by plotting the optimal values $\alpha_t^*$ of the clustering parameter $\alpha$ and the corresponding balls $\mathcal{B}(\alpha_t^*, m^{-\beta})$ used in our definition of task similarity (Figure 2).

The intervals of the parameter induced by these balls correspond to the discretization used by Algorithm 2. We notice a stronger correlation in task similarity for the Gaussian mixture clustering tasks, which implies that meta-learning is more effective here (both in terms of learning test tasks faster, and with lower regret). For knapsack the task similarity is also high, but it turns out that for our dataset there are very 'sharp peaks' at the optima of the total knapsack values as a function of the parameter $\rho$. So even though meta-learning helps us get within a small ball of the optima, a few steps

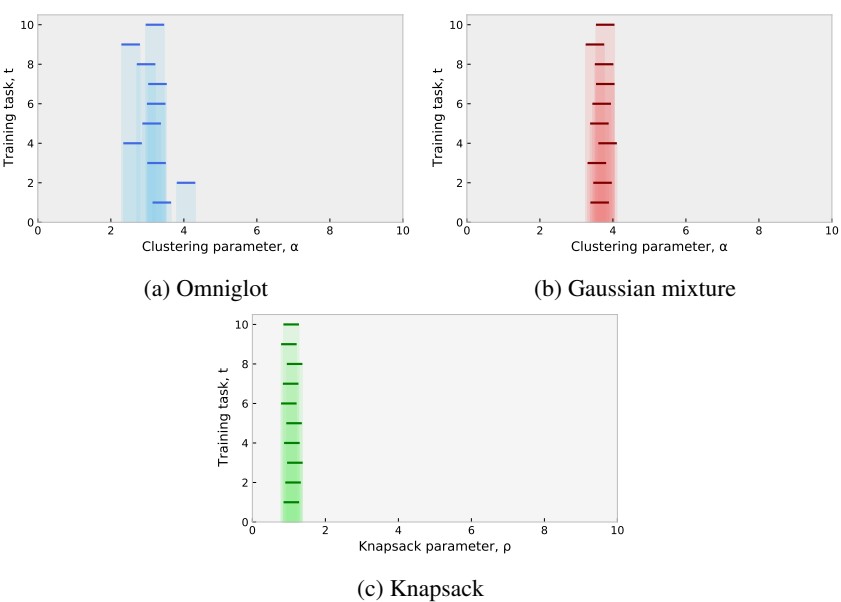

(a) Omniglot        (b) Gaussian mixture

(c) Knapsack

Figure 2: Location of optimal parameter values for the training tasks.

are still needed to converge and we do not see the single-shot benefits of meta-learning as we do for the Gaussian clustering experiment.

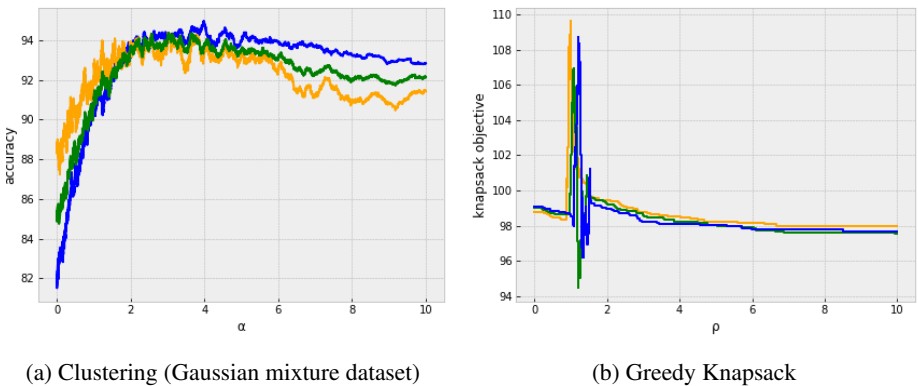

(a) Clustering (Gaussian mixture dataset)        (b) Greedy Knapsack

Figure 3: Average performance (over algorithm randomization) for a few tasks as a function of the configuration parameter. This explains why, despite high task similarity in either case, few-shot meta-learning works better for the Gaussian mixture clustering.