# OpenReview forum: "Learning-to-learn non-convex piecewise-Lipschitz functions"
_NeurIPS.cc/2021/Conference — NeurIPS 2021 Poster_

### Official Review · Reviewer_dmBz · 2021-07-15

**Rating:** 6
**Confidence:** 3

**Summary:**

The paper study the initialization-based meta-learning focus on the piecewise-Lipschitz functions whose discontinuities are dispersed. The authors designed an initialization method that implicitly run on online convex optimization procedure over an adaptive discretization to improve the performance of the exponential forecaster across multi-task. To demonstrate the usefulness of their proposed method, the authors instantiate the method in two settings: online configuration of clustering algorithms and adversarial robustness in online learning.

**Limitations And Societal Impact:**

Yes

**Main Review:**

A good initialization is critical to the success of a meta-learning method. The paper designed the meta-learning of the initialization and step-size of online learning algorithms for piecewise-Lipschitz functions in a multi-task environment. The method optimizes a sequence of data-dependent upper-bounds on the within-task regret. I consider this work interesting, However, I have some concerns too.
1.	To demonstrate the effectiveness of the proposed new meta-initialization algorithm, the authors design experiments for clustering on the real Omniglot dataset and a simulated Gaussian mixture binary classification dataset. As the result showed in the Omniglot dataset, the meta-initialized method has little improvement over single task.. The authors should do more ablation experiments to demonstrate the effective of the proposed meta-initialization method.
2.	In the original experiments, the authors chose the same hyperparameters of meta-initialization method at different experiment situations. The authors should try different hyperparameters to obtain best performance in various tasks.
3.	The authors need to add more experiments to compare other initialization and step-size search methods.
4.	More experiments and datasets are needed to further investigate the robustness and limitations of this approach.
5.	Note the addition of punctuation in some sentences！For example, In Line 44,48-49.



**Time Spent Reviewing:**

72 hours

---

> ### Author Response · Authors · 2021-08-10
> **Response Reviewer dmBz**
>
> Thank you for your positive review. We hope to address your concerns below. In particular we thank the reviewer for recommendations for additional experiments that help further demonstrate empirical usefulness of our algorithms, for which strong theoretical guarantees have been obtained. The experiments (in the main paper, plus new ones  based on the reviewer’s suggestions at this [link](https://drive.google.com/file/d/1-ES8oX5iAjromrBdAYFm9ndTrnP_gVfR/view)) show that our proposed algorithms are able to leverage task similarity in the challenging piecewise Lipschitz setting and show statistically significant improvement over the single task baseline  across multiple datasets.
>
> ### Responses to questions
>
> 1\. [*To demonstrate the effectiveness of the proposed new meta-initialization algorithm, the authors design experiments for clustering on the real Omniglot dataset and a simulated Gaussian mixture binary classification dataset. As the result showed in the Omniglot dataset, the meta-initialized method has little improvement over single task.. The authors should do more ablation experiments to demonstrate the effective of the proposed meta-initialization method.*]
> - Thank you for the suggestion. On the reviewer’s recommendation, in the new experiments we have provided we study two additional datasets, where we also obtain statistically significant improvements using the proposed methods.
>
> 2\. [*In the original experiments, the authors chose the same hyperparameters of meta-initialization method at different experiment situations. The authors should try different hyperparameters to obtain best performance in various tasks.*]
> - Thank you for the suggestion. On the reviewer’s recommendation, in the new experiments we have provided we tuned hyperparameters to obtain the best performance in various tasks.
>
> 3\. [*The authors need to add more experiments to compare other initialization and step-size search methods.*]
> - Unfortunately we are not aware of any other initialization/step-size methods that would work for this setting (known methods in prior literature are gradient-based and do not extend to the piecewise Lipschitz setting). If you have any specific suggestions in mind we would be happy to add them in revision.
>
> 4\. [*More experiments and datasets are needed to further investigate the robustness and limitations of this approach.*]
> - Thank you for the suggestion. In our revised version we will include new experiments applying our approach to multi-task tuning of the greedy algorithm for Knapsack and for center-based clustering of MNIST digits. The specific experimental settings and results are described at this [link](https://drive.google.com/file/d/1-ES8oX5iAjromrBdAYFm9ndTrnP_gVfR/view).
>
> 5\. [*Note the addition of punctuation in some sentences！For example, In Line 44,48-49.*]
> - Thanks, we will fix this.

---

### Official Review · Reviewer_BRfJ · 2021-07-16

**Rating:** 6
**Confidence:** 4

**Summary:**

The authors studied the initialization-based meta-learning of piecewise-Lipschitz functions, demonstrating how online convex optimization over an adaptive discretization can find an initialization that improves the performance of the exponential forecaster across tasks, assuming the tasks have related optima. They then applied this result in two settings: online configuration of clustering algorithms and adversarial robustness in online learning.

**Limitations And Societal Impact:**

I think this paper is theoretically sound. One suggestion is to incorporate more experiments to demonstrate the efficiency of the proposed algorithms.

**Main Review:**

Many existing works of  meta-learning an initialization are largely restricted to the convex Lipschitz setting. This pape relaxes this assumption to study the meta-learning of online algorithms over piecewise-Lipschitz  functions, which can be nonconvex and highly discontinuous. This is the first theoretical study of meta-learning in these application settings.

**Time Spent Reviewing:**

2

---

> ### Author Response · Authors · 2021-08-10
> **Response to Reviewer BRfJ**
>
> Thank you for your positive review. We are not quite sure what type of efficiency you’d hope to see from additional experiments (it could be computational efficiency or efficiency in terms of the number of tasks required). Note that computationally our method adds limited overhead (solving a convex optimization problem) on top of existing efficient within-task algorithms for online learning piecewise-Lipschitz functions.
>
> Following the suggestions by you and Reviewer dmBz of including more experiments, in our revised version we will include new experiments applying our approach to multi-task tuning of the greedy algorithm for Knapsack (on a synthetic dataset) and for center-based clustering of MNIST digits (same clustering setting as Omniglot and Gaussian mixtures in the paper). The experiments show that our proposed algorithms are able to leverage task similarity in the challenging piecewise Lipschitz setting and show statistically significant improvement over the single task baseline  across multiple datasets. For more details please our response to Reviewer dmBz and results at the following [link](https://drive.google.com/file/d/1-ES8oX5iAjromrBdAYFm9ndTrnP_gVfR/view).

---

### Official Review · Reviewer_4fEk · 2021-07-23

**Rating:** 7
**Confidence:** 3

**Summary:**

This paper proposed a meta-learning approach to learn piecewise-Lipschitz functions in the multi-task setting. Simply applying a traditional single-task exponential forecasting method multiple times cannot make the regret bound diminishes as the number of tasks $T$ increases. To tackle this issue, they first attempt to obtain a bound that involves the task similarity term, and then the authors propose two meta-learning algorithms, based on the observation that the regret bound depends on two things: learning rate and initialization probability choosing. Hence, they proposed two meta-learning algorithms, to efficiently learn to choose better initialization and learning rate. The theoretical results show that this algorithm can provably improve the baseline of applying a single-task algorithm $T$ times.


**Limitations And Societal Impact:**

I notice that in Line 214:  with positive volume $vol(C_t) > 0$ that is revealed after choosing $w_t$. I think $C_t$ is the ball center at optimum at $\rho_t^*$, so does it mean we need some oracle knowledge to reveal $vol(C_t)$?


**Main Review:**

This paper proposes a novel meta-learning approach for online learning of piecewise-Lipschitz functions   in a multi-task environment. To my best knowledge, this paper is the first to generalize learning piecewise-Lipschitz functions problem to multi-task setting. It shows that the trivially applying single-task algorithm $T$ times will not make regret bound vanish, and then naturally introduce a quantity $V$ that captures the task similarity. Based on the bound, the proposal of two meta algorithms to learn to choose hyperparameters is also novel to me. The theoretical results look sound as far as I checked. The paper is well-organized and the motivations are very clear. I do not have much experience in this field but I learnt from this paper and enjoy reading it. Hyperparameter choosing is a longstanding difficulty in the machine learning community, and this paper sheds some light on this aspect, and the present paper has some merits.


**Time Spent Reviewing:**

3

---

> ### Author Response · Authors · 2021-08-10
> **Response to Reviewer 4fEk**
>
> Thank you for your positive review. To answer your question: yes, we assume oracle knowledge of $\rho_t^\ast$, the optimum-in-hindsight for tasks $t$, which reveals $C_t$. We state this assumption on line 114. Note that for the typical applications of our work this is a reasonable assumption; for example, past analyses [10] study the effectiveness of empirical risk minimization in the statistical algorithm design setting, which involves the same type of optimization problem.

---

### Public Comment · Authors · 2023-08-30
**Consult version history for code**

https://openreview.net/revisions?id=USq7LP5pnDH

---

### Decision · Program_Chairs · 2021-09-27

**Decision:**

Accept (Poster)

**Comment:**

The paper gives an online algorithm for learning convex piecewise functions under certain assumptions regarding dispersion of the points of discontinuity. This follows recent work on algorithm meta-design and online learning.